

# Understanding the role of water and tillage erosion from $^{239+240}$Pu tracer measurements using inverse modelling

Florian Wilken[1,2], Michael Ketterer[3], Sylvia Koszinski[4], Michael Sommer[4,5] and Peter Fiener[2]

[1]Department of Environmental Systems Science, Eidgenössische Technische Hochschule Zürich, Zürich, Switzerland
[2]Institute for Geography, Universität Augsburg, Augsburg, Germany
[3]Chemistry and Biochemistry, Northern Arizona University, Flagstaff, USA
[4]Working Group Landscape Pedology, Leibniz-Centre for Agricultural Landscape Research ZALF e.V., Müncheberg, Germany
[5]Institute of Earth and Environmental Sciences, University of Potsdam, Potsdam, Germany

*Correspondence to:* Florian Wilken (florian.wilken@usys.ethz.ch)

## Abstract

Soil redistribution on arable land is a major threat for a sustainable use of soil resources. The soil redistribution process most studies focus on is water erosion, while wind and tillage erosion also induce pronounced redistribution of soil materials. Especially, tillage erosion is understudied, as it does not lead to visible off-site damages. The analysis of on-site / in-field soil redistribution is mostly based on tracer studies, whereas radionuclide tracers (e.g. $^{137}$Cs, $^{239+240}$Pu) from nuclear weapon tests are commonly used to derive the erosion history over the past 50-60 yr. Tracer studies allow to determine soil redistribution patterns, but integrate all kinds of soil redistribution processes and hence do not allow to unravel the contribution of different erosion processes. The aim of this study is to understand the contribution of water and tillage erosion leading to soil patterns found in a small hummocky ground moraine catchment under intensive agricultural use. Therefore, $^{239+240}$Pu derived soil redistribution patterns were analysed using an inverse modelling approach accounting for water and tillage erosion processes. The results of this analysis clearly point out that tillage erosion is the dominant process of soil redistribution in the small catchment, which also affects the hydrological and sedimentological connectivity between arable land and the kettle hole. A topographic change up to 17 cm (53 yr)$^{-1}$ in the eroded parts of the catchment is not able to explain the current soil profile truncation that exceeds the $^{239+240}$Pu derived topographic change substantially. Hence, tillage erosion is not limited to the time since the onset of intense mechanisation since the 1960s. In general, the study stresses the urgent need to consider tillage erosion as a very important soil degradation process that drives patterns of soil properties in our arable landscapes.



## 1. Introduction

Soil erosion is a major threat for the supply of soil related ecosystem services (Montanarella et al., 2016). Over the past decades, particularly the off-site effects associated with water erosion, like nutrient inputs from arable lands into inland waters (Pimentel and Burgess, 2013) or siltation of reservoirs (Krasa et al., 2019), were in the scientific and political focus. Within the European Union, the focus on off-site erosion effects is partly caused by the definition of the goals of the EU Water Framework Directive (EU 2000/60/ES) that focuses mainly on water bodies and floodplains. Thereby, other soil erosion drivers than water are somewhat out of the scope of most studies. Tillage erosion is a mostly ignored soil erosion process (Fiener et al., 2018) that, however, substantially contributes to on-site effects on soil properties and hence agricultural productivity (Winnige, 2004; Nie et al., 2019). Van Oost et al. (2006) pointed out that tillage erosion rates are globally at least in the same order of magnitude as water erosion rates. Particularly areas of a hilly topography with short summit-footslope distances, such as young morainic areas, can be subject to pronounced in-field soil degradation patterns caused by tillage erosion (Winnige, 2004; Deumlich et al., 2017). Young morainic areas that are under intense arable cultivation and corresponding tillage erosion are widespread in northern Europe, Canada, northern USA, Russia and eastern Argentina.

Measuring or monitoring water and tillage erosion is challenging as both processes are interlinked and correspond to topography (Van Oost et al., 2005b; Van Oost et al., 2006). Water erosion calls for a sufficiently long monitoring time (typically decades) to cover a statistically representative variation of rainfall events occurring on different land cover conditions (Fiener et al., 2019). Therefore, thousands of plot experiments, either driven by natural or artificial rainfall simulations, were carried out in different environments and different land cover conditions (Cerdan et al., 2010; Auerswald et al., 2014). Furthermore, a large number of small catchment studies were performed to include both erosion and depositional processes (for overview see; Fiener et al., 2019). However, soil erosion monitoring is mostly based on sediment delivery outlet monitoring, which cannot address internal redistribution. In contrast, tillage erosion can only be measured based on the movement of beforehand applied tracers or by morphological change monitoring (for an overview of tillage erosion measuring techniques see; Fiener et al., 2018). However, these beforehand applied tracers and change monitoring methods cannot assess the soil redistribution history of sites. Natural or anthropogenic tracers that are inherent to specific soil layer characteristics can be used to understand the soil redistribution (Fiener et al., 2018). Especially anthropogenic radionuclides (e.g. $^{137}$Cs, $^{239+240}$Pu, $^{210}$Pb, $^{7}$Be) have demonstrated their ability to determine changes in topography (Parsons and Foster, 2011; Mabit et al., 2014; Alewell et al., 2017; Deumlich et al., 2017). Nuclear weapon tests lifted radioisotopes up to the stratosphere, where mixture led to an almost homogeneous distribution and corresponding fallout by rain (Alewell et al., 2017). The main period of atmospheric nuclear weapon tests is from 1953 to 1964 (Schimmack et al., 2001), while the limited test ban treaty caused a rapid end of atmospheric bomb tests in 1963-1964 (Wallbrink and Murray, 1993). This rapid end leads to a distinct peak in the activity of radioisotopes in soils, which enables the use of radioisotopes as redistribution tracers in soils. The radioisotope $^{137}$Cs has been used as a soil redistribution tracer in a large number of studies (e.g. Van Oost and Govers, 2006; Porto and Walling, 2012; Zhang, 2015; Greenwood and Meusburger, 2019; Srivastava et al., 2019) and has become a widely used method



in soil erosion science. However, the Chernobyl disaster (1986) re-contaminated large areas of Europe with $^{137}$Cs. An unmixing of the Chernobyl fallout signal from the original (1960s) bomb peak was able by the use of the $^{134}$Cs/$^{137}$Cs ratio (Lust and
Realo, 2012). However, due to the short $^{134}$Cs half-life of 2 yr (Schimmack et al., 2001), this method cannot be applied anymore. Hence, the use of $^{137}$Cs as a soil redistribution tracer in Europe is associated with large uncertainties. Furthermore, due to the $^{137}$Cs half-life of about 30 yr, decay has already been led to a pronounced reduction of the activity until today (Alewell et al., 2017). Over the past decade, $^{239+240}$Pu has been discussed and tested as an alternative radioisotopic tracer for soil erosion studies. Decay is not an issue as the half-life of $^{239+240}$Pu is long ($^{239}$Pu = 24000 yr; $^{240}$Pu = 6563 yr.) and the
$^{239+240}$Pu contamination by the Chernobyl accident was spatially very limited and can be determined by the $^{239}$Pu/$^{240}$Pu ratio (Alewell et al., 2014; Alewell et al., 2017).

Radionuclide tracers integrate soil erosion processes over time (e.g. since the bomb peak of the 1960s in case of $^{137}$Cs and $^{239+240}$Pu), which somewhat averages out the large temporal variability of water (episodic nature) and tillage (mechanisation) erosion. However, the use of radioisotopic tracers integrates all kinds of soil redistribution processes and lacks insights on the
relative contribution of the driving processes at play. To unravel these different processes calls for an inverse modelling approach carrying out model runs of different parameterisations to alter the contribution and mechanisms of different soil redistribution drivers. There are only few models fusing both water and tillage erosion. Physically oriented models like MCST-C (Wilken et al., 2017b) and LandSoil (Ciampalini et al., 2012) work event based and are developed for process understanding, while conceptual USLE based models (WaTEM/SEDEM: Van Oost et al., 2000; Van Rompaey et al., 2001)
aim at a robust prediction of long-term soil erosion rates (Alewell et al., 2019). For an inverse modelling approach to unravel tillage and water erosion based on radionuclide tracer, it is necessary to use a parsimonious approach with a limited parameter space and available input data over five to six decades, which suggests the use of conceptual models.

In this study, we will determine the soil redistribution patterns in a small 4.2 ha catchment based on high-resolution $^{239+240}$Pu measurements and analyse the contribution of water and tillage erosion processes based on an inverse modelling approach
using a combined water and tillage erosion model. The general aim is to unravel the importance of water and tillage erosion for current soil properties in an intensively used arable landscape of northeastern Germany.

## 2. Methods

### 2.1 Study area

The study area (53°21'2 N, 13°39'5 E) is situated in the hummocky ground morainic landscape of the Weichselian glacial belt
('young morainic area') of northeastern Germany (Fig. 1). Characteristic for these landscapes are widespread closed depressions, so-called kettle holes, which result from a delayed melting of dead ice blocks. They are nowadays filled with mineral soil, (degraded) peat or water. The study area is part of a kettle hole catchment (4.2 ha; Fig. 1) and consists of different topographical locations. The recent crop rotation is rape (*Brassica napus* L.)-winter wheat (*Triticum aestivum* L.)-winter barley (*Hordeum vulgare* L.)-winter barley, cultivated without cover crops, which is a typical conventional crop rotation that is



adapted for the highly fertile soils of the Uckermark region. The mean arable land of a farm in the region is 352 ha, which is much larger compared to the mean of the State of Brandenburg (250 ha) and Germany (60 ha; Troegel and Schulz, 2018). These larger field sizes are mainly caused by land consolidation in the 1960s during the socialist period. However, also before, agriculture in the region was already characterised by large scale farming structures (Wolz, 2013) and corresponding high agricultural mechanisation. The catchment is part of a single large field (54 ha), which is a size that can be frequently found

in the region. The soils are developed from glacial till and vary with respect to their location in the landscape. Convex hilltops and steep slopes are dominated by extremely eroded A-C profiles (Calcaric Regosols, soil classification according to: IUSS, 2015), while Luvisols showing different degrees of erosion and are typically situated at the up and mid slopes, the footslopes and depressions show a sequence of Gleyic-Colluvic Regosols, Mollic Gleysols and (buried) Terric Histosols (Sommer et al., 2008; Gerke et al., 2010). The closed kettle hole depression itself is built up by degraded Histosols and covered by a thin

colluvial layer of mineral soil (40±8 cm mean, 25% and 75% quantile, n: 20; based on soil auger prospection). The study area has a continental climate (Köppen: Dfb) with low annual precipitation (500 mm; Fiener et al., 2018) and high temperature amplitude (Jul: 18°C, Jan: 0°C; mean 8.9°C WMO-CLINO 1981-2010 for the meteorological stations Gruenow and Angermuende). Between 7 and 11 erosive rainfall events per year take place (Deumlich, 1999). In the region, maximum intensities up to 162 mm h$^{-1}$ (10-min interval) were recorded during an extreme event in 2016 (Wilken et al., 2018).

**2.2 Soil sampling design and preparation**

The soil sampling design was organised in a regular 20 m x 20 m grid with at least one sampling point exceeding the spatial extent of the catchment under study (Fig. 1). To assess small scale spatial variability (for distances of sampling points between 5 m and 20 m) a nested sampling approach was applied (Fig. 1; Hengl and MacMillan, 2019). Therefore, five densified sub-grids, located at different topographical locations (hilltop, ridge shoulder, moderately steep mid slope, steep mid slope, foot

slope/valley; Fig. 1) were selected. In total, 209 locations were included in the sampling design.

The sampling points were located using a differential GPS (AgGPS$^{TM}$ 132; Trimble Navigation Ltd, Sunnyvale CA, USA) applying the SAPOS (LGB) correction signal. With respect to drill penetrability, soil sampling was carried out under moderately wet soil moisture conditions (range from 18% to 32% and 16% to 33% for topsoil and subsoil, respectively) in December 2015. Closed soil cores, using a steel drill containing a plastic liner (4.6 cm inner diameter), were driven by a

percussion corer (Cobra TTe; Atlas Copco Power Techniques GmbH, Stockholm, Sweden) into the ground. Soil cores driven down to a depth of 50 cm were taken at all 209 sampling points. To account for the vertical distribution of $^{239+240}$Pu in the soil profile, ten replicate soil cores were taken at five up-hill and five foot-slope locations (Fig. 1; P1-P10). Thus, overall 219 soil cores at 209 locations were taken (Fig. 1).

To minimize physical and biogeochemical disturbance, the soil cores were stored in a freezer until sample preparation. After

complete thawing, the soil cores were separated into topsoil (Ap horizon) and subsoil. The separation was done by visual interpretation of soil horizon characteristics (colour, structural and density differences), showing a variation in topsoil thickness between 16 and 30 cm (mean of 23.5 cm) depending on the topographic position. Aliquot (n: 3) samples for each topsoil and



subsoil location were taken for gravimetric water content (weighted before and after drying at 105°C) and dry bulk density (known sample volume) measurements. The soil samples were air dried and subsequently sieved with a 2 mm mesh to separate

stones from the fine soil.

## 2.3 $^{239+240}$Pu measurements

All topsoil samples (n: 209) were analysed, while 145 (~70 %) of the subsoil samples of highest topsoil activities were measured on their $^{239+240}$Pu activity. This was done to reduce the inventory of samples that have Pu activities below the detection limit. Furthermore, at 10 locations high-resolution (5 cm) depth increments were measured to assess the depth

distribution of $^{239+240}$Pu activity below the mixed plough layer.

The fallout radionuclides $^{239+240}$Pu were used to estimate effective soil redistribution since the 1960s. Plutonium isotopes measurements were conducted following Calitri et al. (2019) based on the procedure of Ketterer et al. (2004). Before the mass spectrometry analysis, 10 g of milled fine earth were dry-ashed for at least 8 hours at 600°C to remove organic matter. Subsequently, the samples were spiked using 30 pg (c. 0.0044 Bq) of a $^{242}$Pu tracer solution (NIST 4334). The sample leaching

was applied using 16 M nitric acid (HNO$_3$) overnight at 80°C and subsequently filtered and adjusted to a concentration of 8 M HNO$_3$. Plutonium species were adjusted to the Pu (IV) oxidation state using first an acidified FeSO$_4$·7H$_2$O solution (2 mg ml$^{-1}$ of leached solution) and subsequently a sodium nitrite (NaNO$_2$) solution (20 mg ml$^{-1}$ of leached solution). The samples were heated at 75°C for two hours. Tetravalent Pu was separated from the leached solution using a Pu-selective TEVA resin (2 mg of TEVA per millilitre of leached solution). Following occasional agitation for two hours, the resin was collected in a

pipette tip equipped with a glasswool plug. This disposable column was rinsed with 2 M aqueous HNO$_3$ to remove unretained matrix elements [e.g. uranium (U)], then rinsed with 8 M HCl to elute thorium (Th) and finally rinsed again with 2 M aqueous HNO$_3$ (rinse volume: 1 ml per 30 mg of TEVA). Plutonium was eluted using 0.05 M aqueous ammonium oxalate. Data quality was evaluated through the analysis of blanks (soils or rocks devoid of Pu), duplicates and control samples of known $^{239+240}$Pu activities (Standard Reference material 4350b – River sediment for radioactivity measurements from NIST). Activities of

$^{239+240}$Pu were measured using a Thermo X Series II quadrupole ICPMS, located at Northern Arizona University. The ICPMS instrument is equipped with an APEX HF high-efficiency sample introduction system. The masses of $^{239}$Pu and $^{240}$Pu in the samples were converted into the summed activity $^{239+240}$Pu.

## 2.4 $^{239+240}$Pu based soil erosion assessment

We applied the proportional conversion approach of Walling et al. (2011). Erosion is calculated following equation 1 and 2:

$$SL_i = 10 * BD_i * TD_i * RR_i * T^{-1} \tag{1}$$

where $SL_i$ is the mean annual soil loss in Mg ha$^{-1}$ yr$^{-1}$, $BD_i$ is the soil bulk density in kg m$^{-3}$, $TD_i$ is the vertical depth of the Ap horizon in m (tillage depth), $RR_i$ is the relative reduction of the reference inventory of the $^{239+240}$Pu inventory and $T$ is the years that are elapsed since the end of atmospheric nuclear weapon tests (mainly since 1964).



$$RR_i = \frac{(Pu_{ref} - Pu_i)}{Pu_{ref}} \qquad (2)$$

where $Pu_i$ is the inventory at sampling point $i$ and $Pu_{ref}$ is the reference inventory of undisturbed sites in Bq m[-2].

For a three-dimensional representation of the [239+240]Pu redistribution by water and tillage erosion, SPEROS-Pu was developed

that is based on a modified version of the SPEROS-C model (Van Oost et al., 2005a; Fiener et al., 2015; Nadeu et al., 2015). We utilized the water and tillage erosion modules of the SPEROS model that are based on WaTEM/SEDEM (Van Oost et al., 2000; Van Rompaey et al., 2001). SPEROS-Pu is a spatially explicit water and tillage erosion model. Water erosion is simulated based on a gridded application of the Revised Universal Soil Loss Equation (RUSLE: Renard et al., 1996) coupled with a sediment transport and deposition approach. Erosion is calculated according to a slightly modified RUSLE approach,

transport and deposition is based on the grid cell specific local transport capacity $TC$ (kg m[-1] yr[-1]), which multiplies RUSLE factors with a transport capacity coefficient ($ktc$; in m)

$$TC = ktc * R * C * K * L * S * P \qquad (3)$$

where $R$, $C$, $K$, $L$, $S$ and $P$ are the RUSLE factors (see Renard et al., 1996).

The tillage erosion module of SPEROS-C follows a diffusion-type equation adopted from Govers et al. (1994) that derives tillage erosion based on change in topography and management specific coefficients

$$Q_{til} = -k_{til}\frac{\Delta h}{\Delta x} \qquad (4)$$

where $Q_{til}$ is the soil flux in kg m[-2] yr[-1], $\Delta h$ is the elevation difference in m, $\Delta x$ is the horizontal distance in m while $k_{til}$ is the tillage transport coefficient in kg m[-1] yr[-1]

$$k_{til} = BD_i * TD_i * x_{til} \qquad (5)$$

where $x_{til}$ is the tillage translocation distance in m (for $BD_i$ and $TD_i$ see description of eq. 1).

The representation of the [239+240]Pu redistribution in the SPEROS-Pu model is three dimensional and accounts for [239+240]Pu source area depletion and corresponding redistribution of depleted sediments. The horizontal distribution of [239+240]Pu is grid

based, while the vertical distribution is represented by ten 10 cm layers. The two uppermost layers are assumed to be homogeneously mixed by tillage operations and have the same average [239+240]Pu activity of the upper two soil layers. At the beginning of the simulation, the [239+240]Pu reference activity is homogeneously distributed within the mixed plough layer and over the entire catchment. Subsequently, the local [239+240]Pu inventory is altered by soil redistribution processes. Soil erosion processes lead to a reduction of the [239+240]Pu inventory per m[2] due to a soil and corresponding [239+240]Pu loss and mixing in of

non-contaminated subsoil. Deposition adds contaminated material on top and increases the [239+240]Pu inventory.

### 2.5 Implementation and inverse modelling

### Soil redistribution based on [239+240]Pu measurements

Spatially distributed top and subsoil bulk density and tillage depth information for each individual sampling location were applied to the proportional conversion approach. The reference inventory of undisturbed sites follows the value determined by





Calitri et al. (2019) who found a $^{239+240}$Pu inventory of 43±3 Bq m² at a location 8.5 km apart from the study area. To account

for the uncertainty inherent to the reference determined in close proximity but not at site, a reference range from 40 to 46 Bq

m² was accounted for.

The point data of the $^{239+240}$Pu inventory for the depth 0-50 cm was geostatistically (block kriging) interpolated for a gridded

spatial representation that matches the spatial resolution of the soil redistribution model (5 m x 5 m). Block kriging was used

to reduce small-scale scattering that is naturally inherent to soil cores of 4.6 cm diameter that are supposed to represent

decametre scale. Different block sizes were tested in the kriging approach and finally 20 m selected that matches the sampling

resolution and has shown not to lead in over-smoothening the interpolation result. The theoretical semivariogram model (Fig.

2) was fitted using all 209 sampling points, including the nested samples to account for variations over short distances.

However, the input data of the interpolation itself solely uses the regular 20 m x 20 m grid points. The interpolation and

geostatistical analysis were carried out using the statistical software GNU R (R-Core-Team, 2015), version 3.5.3 and the add-

on package gstat (Pebesma, 2004).

**Inverse modelling of water and tillage erosion**

An inverse modelling approach was used to understand the proportion of water and tillage erosion that is inherent to the

$^{239+240}$Pu based soil erosion map. The inverse modelling iterates three parameter sets over the 53 yr. modelling period (1964-

2016): (i) the $k_{til}$ tillage translocation coefficient, (ii) the product of all RUSLE factors (as given in Eq. 3; in the following

referred to water erosion strength) and (iii) the water erosion transport capacity coefficient $ktc$ (see Table 1) that controls the

transport distance and is the standard calibration parameter of the model. While changes of the tillage translocation coefficient

and water erosion strength only alter the quantity of soil redistribution, the $ktc$ has a pronounced impact on spatial patterns of

modelled soil redistribution. The parameter range covers very low to extreme soil redistribution rates ($k_{til}$ max: 1000 kg m⁻¹;

$ktc$ max: 500 m; RUSLE-factors product deviation: 100%; see Table 1). The water erosion reference parameterisation for $ktc$

is 150 m (Van Oost et al., 2003). The interplay of parameter combinations was assessed in 35722 different model runs (Table

1).

To determine the model match, different goodness of fit parameters were calculated that compare the interpolated $^{239+240}$Pu

raster map against the results of the raster results calculated by the inverse modelling approach. To address the high spatial

variability of the 5 x 5 m raster by raster comparison, a classification of results was carried out. Therefore, mean values were

calculated based on the cells that fall into a specific class of the interpolated map of $^{239+240}$Pu inventories. The classification

covers 20 classes with 2.5 Bq m⁻² steps from 17.5 to 65 (and a class >65) Bq m⁻² of $^{239+240}$Pu. First, the spatial correlation was

calculated for both the raster by raster comparison and the classified results. Second, the classified results of the inverse

modelling were tested using goodness of fit parameters that take absolute differences between observed and predicted values

into account (RMSE, model efficiency coefficient: MEF according to Nash and Sutcliffe, 1970). As a last step, the results of

the measured and modelled $^{239+240}$Pu inventories were transferred into tillage and water erosion maps (given in Mg ha⁻¹ and





topographical change in cm) applying the proportional conversion approach (see chapter $^{239+240}$Pu based soil erosion assessment).

## 3 Results

### 3.1 $^{239+240}$Pu activities and inventories

The topsoil and subsoil $^{239+240}$Pu activities differ substantially from each other and show distinct spatial patterns according to their topographic position. All topsoil samples (n: 209) showed a $^{239+240}$Pu activity above the detection limit, while 7 of the 145 subsoil samples fall below the detection limit (< 0.002 Bq kg$^{-1}$). The average $^{239+240}$Pu activity is 0.078 ± 0.016 Bq kg$^{-1}$ and 0.035±0.038 Bq kg$^{-1}$ for topsoil and subsoil, respectively. The high-resolution depth profiles (5 cm increments) distinctively show a sharp reduction of the $^{239+240}$Pu activity below the plough layer at erosional sites (Fig. 3). In turn, depositional sites show more complex depth distributions. Location P4 (Fig. 3) does not show a drop in activity until a depth of 0.5 m, while P1 (and partly P3) shows an increase of the $^{239+240}$Pu activity with depth that potentially goes back to Pu enrichment processes during lateral transport or the deposition of already Pu depleted source material into the topsoil. The topsoil $^{239+240}$Pu/reference (43 Bq m$^{-2}$) ratio indicates soil erosion related $^{239+240}$Pu depletion or enrichment according to a ratio lower and higher than one, respectively. The highest depletion (min: 0.28; 5% quantile: 0.37) can be found at the hilltops that are most affected by tillage, while enrichment (max: 1.18; 95% quantile: 0.92) is spatially limited to the flat surroundings of the kettle hole (Fig. 1), where topsoil material was potentially deposited by both redistribution processes. At 14 (of 209) sampling locations, a higher subsoil than topsoil $^{239+240}$Pu activity was found. The majority of these locations show enriched $^{239+240}$Pu activities in the subsoil (9 of 14) that goes in line with higher subsoil activities compared to the topsoil (13 of 14), which is caused by deposition of $^{239+240}$Pu depleted sediments. These locations are all, except for one, located at the kettle hole surrounding plateau where both water and tillage erosion show deposition. The $^{239+240}$Pu depletion of sampling points (including locations outside the study catchment) goes down to 12 Bq m$^{-2}$ while the highest five locations exceed 86 Bq m$^{-2}$, which means that the $^{239+240}$Pu inventory has more than doubled compared to the reference inventory of 43 Bq m$^{-2}$ and therefore a tillage depth of 25 cm, which has to be attributed to enrichment processes. Therefore, these five locations were excluded from the analysis.

The distribution of the interpolated maps covering the study catchment shows that substantially more locations fall below the reference inventory than exceeding it (Fig. 4). This indicates that a larger area is subject to erosion processes compared to depositional processes.

### 3.2 $^{239+240}$Pu measurements vs. inverse water and tillage erosion modelling

To understand the drivers of current $^{239+240}$Pu and associated soil degradation patterns, an inverse modelling was carried out that was qualitatively analysed by goodness of fit parameters. The spatial correlation between the $^{239+240}$Pu derived patterns and the modelled soil erosion, including both water and tillage erosion, is only moderate (R$^2$: 0.45) on a raster by raster





comparison (n: 1699, 5 m x 5 m grid points; Fig. 5a). This is not surprising due to short term and small scale topographic dynamics (e.g. roughness, wheel tracks, vegetation artefacts) that transfer into model results as a static digital elevation model

is used. To reduce this small-scale variability and understand the average goodness of fit, the inverse modelling results were classified according to the measured $^{239+240}$Pu activity. The classified results average out the spatio-temporal dynamics and show a very high correlation ($R^2$: 0.95, Rho: 0.99; Fig. 5b), which pinpoints great agreement of the spatial soil redistribution patterns between the $^{239+240}$Pu measurements and the model results.

While the analysis of the spatial correlation is a relative comparison, the absolute deviation is taken into account according to

the MEF (model efficiency coefficient; Nash and Sutcliffe, 1970; 1 = perfect prediction, 0 = as good as mean of all measurements, < 0 = worse than mean). The quality of model predictions shows hardly any sensitivity to water erosion related parameterisations ($ktc$ and erosion strength; Figure 6c). In contrast, the tillage erosion strength, represented by $k_{til}$ parameter iterations, showed a substantial impact on the MEF (Fig. 6a & b). The best model fit was found for a $k_{til}$ of 350 kg m$^{-1}$ achieving a MEF of 0.87, while the corresponding RMSE is 5.2 Bq m$^{-2}$. The best model fit was found without the contribution

of water erosion.

Soil redistribution determined by the proportional conversion approach using $^{239+240}$Pu measurements, indicates substantial morphodynamics in the study catchment over the past decades. Soil erosion at hilltop locations is shown to be up to 14.9 cm (43 Bq m$^{-2}$ reference; 40 Bq m$^{-2}$ reference: 14.1 cm; 46 Bq m$^{-2}$ reference: 15.6 cm), while deposition partly builds up to 21.5 cm (43 Bq m$^{-2}$ reference; 40 Bq m$^{-2}$ reference: 24.9 cm; 46 Bq m$^{-2}$ reference: 18.6 cm) colluvium over the past 53 yr. The inverse

modelling stresses that substantial soil erosion, which takes place over large areas, is almost exclusively attributed to tillage translocation (modelled max. water erosion: 3.8 cm (53 yr)$^{-1}$ vs. max. tillage erosion: 13.5 cm (53 yr)$^{-1}$; Fig. 7c, d). In turn, both processes contribute to deposition in the kettle hole surrounding flats (max. water deposition: 27.1 cm (53 yr)$^{-1}$ vs. max. tillage deposition: 22.4 cm (53 yr)$^{-1}$; Fig. 7c, d).

## 4 Discussions

### 4.1 $^{239+240}$Pu methodological limitations

$^{239+240}$Pu has demonstrated its suitability to determine the recent soil redistribution history (since the 1960s; see review Alewell et al., 2017) and fills the gap of upcoming limitations in the use of $^{137}$Cs as a soil redistribution tracer (Parsons and Foster, 2011; Mabit et al., 2013). However, some methodological limitations persist that need to be taken into account. Enrichment processes, due to selective transport of soil constituents that fallout radionuclides are preferentially associated with, are a

critical issue for the use of most (e.g., $^{239+240}$Pu, $^{137}$Cs, $^{210}$Pb) radionuclide tracers (Parsons and Foster, 2011; Mabit et al., 2014; Alewell et al., 2017). While $^{137}$Cs is mainly associated with clay particles, $^{239+240}$Pu binds to soil organic matter and oxides (Alewell et al., 2017) that are less affected by selective water transport and corresponding $^{239+240}$Pu enrichment (Meusburger et al., 2016; Xu et al., 2017). Nevertheless, the $^{239+240}$Pu activity at depositional sites, that are redistributed by water (transport by tillage is typically assumed to be non-grain size selective; Fiener et al., 2018), can be higher in relation to the activity of the





source material. A soil profile that shows a distinct indicator of enrichment processes in this study is sampling profile P1 (25-45 cm; Fig. 3) that is situated in the kettle hole surrounding flat. Hence, enrichment is to some extent also an issue within this study that causes an overestimation of deposition. A particle size correction factor was not applied as topsoil enrichment (topsoil Bq m$^{-2}$ > ref. 43 Bq m$^{-2}$) was exclusively found at very few sampling locations (<6%) in the kettle hole surrounding flats. Furthermore, the mean topsoil ratio of enriched sediments is moderate (1.2) and supports the general assumption that $^{239+240}$Pu is less affected by selective transport compared to $^{137}$Cs (Alewell et al., 2017) and that transport by tillage is non-grain size specific. The counteracting process of enrichment is the deposition of $^{239+240}$Pu depleted sediments that are transported from highly eroded locations. Such highly depleted locations can be found at the hilltops of the study area (Fig. 7b). Hence, the hilltops are the main source of highly depleted sediments that are deposited in kettle hole surrounding flats. However, the minimum horizontal distance from the hilltops to the kettle hole surrounding flat is roughly about 70 m and the approximate tillage translocation distance 0.5 to 1 m per pass (Fiener et al., 2018). Hence, deposition of depleted $^{239+240}$Pu material has to be mainly attributed to surface runoff that can bridge longer transport distances. SPEROS-Pu takes depletion of deposited sediments into account but does not address enrichment processes. Furthermore, a maximum soil sampling depth down to 50 cm was carried out within this study that technically allows to derive a maximum depositional depth of 25 cm using the proportional conversion approach of Walling et al. (2011), which is at four sampling locations exceeded. Nevertheless, also with a deeper soil sampling, it would be arguable if these potentially enriched or depleted sampling locations should be excluded from the statistical analysis, like it was done for extreme depositional locations (4 sampling locations) within this study.

## 4.2 Using $^{239+240}$Pu and inverse modelling to understand the recent soil erosion history

Within the intensively used study catchment, substantial $^{239+240}$Pu derived soil redistribution was found with soil loss up to 45 Mg ha$^{-1}$ yr$^{-1}$ (ref. 43 Bq m$^{-2}$; ref. 40 and 46 Bq m$^{-2}$: 43 and 47 Mg ha$^{-1}$ yr$^{-1}$) and sediment deposition up to 65 Mg ha$^{-1}$ yr$^{-1}$ (ref. 43 Bq m$^{-2}$; ref. 40 and 46 Bq m$^{-2}$: 75 and 56 Mg ha$^{-1}$ yr$^{-1}$). Very high deposition can only be found in the spatially narrow area of the kettle hole surrounding flat where both processes water and tillage erosion lead to deposition (Fig. 1). The kettle hole surrounding flat is a spatially narrow area, but the only region where water erosion substantially contributes to pronounced morphodynamics (Fig. 7d). As a result of the small spatial contribution, the inverse modelling shows hardly any sensitivity on goodness of fit changes in reaction to the variation in model parameterisations (Fig. 6c). Nevertheless, sediment deposition and delivery by surface runoff is a relevant process in the study area. Evidence for runoff based sediment delivery is a colluvial layer covering the peat in the kettle hole with an average depth of 40 cm. This sediment delivery into the kettle hole cannot be explained by the inverse modelling of water erosion applying a reasonable parameter range of the inverse modelling. Hence, the reference parameterisation of the region needs to be assumed as the most appropriate ($ktc$: 150 m and RUSLE parameters according to Tab. 1). According to the model run using the reference parameterisation for water erosion, a colluvial layer of 1.7 cm (53 yr)$^{-1}$ would have been developed on top of the peat over the past decades. This indicates a long water erosion history before the 1960s. This is not surprising as bare soil conditions and erosive rainfall events have taken place since the onset of





arable use approx. 1k yr. before present (Van der Meij et al., 2017; Kappler et al., 2018). In turn, tillage erosion is typically assumed to be a process that is linked to recent developments of increasing mechanical forces that are applied to soils over the

past century (Sommer et al., 2008; Calitri et al., 2019). Within this study, a maximum topographical change by hilltop erosion up to 17 cm (53 yr)$^{-1}$ was determined. In a review on tillage erosion by Van Oost and Govers (2006), tillage translocation coefficients of 44 experiments reported a mean: 234 kg m$^{-1}$; 5% percentile: 30 kg m$^{-1}$; 95% percentile: 640 kg m$^{-1}$ for mouldboard and chisel plough. Within this study, a tillage translocation coefficient of 350 kg m$^{-1}$ per year was determined. The tillage translocation coefficients, determined by the inverse modelling approach, are rather high compared to other studies

considering that fallout radioisotopic tracer approaches integrate a phase of high mechanical development from low to high power farming machines (Sommer et al., 2008; Keller et al., 2019). Although recent tillage translocation rates are rather high they cannot explain the soil depth patterns that are visible by augerings. In the study region, it can be observed that tillage erosion mainly affected hilltops. Calcaric glacial till is approaching the surface by soil profile truncation and partially mixed into the plough layer. Within the study catchment, this is the case for 20 sampling locations (CaCO$_3$ > 0.5%) that are also

indicated as strongest eroded sites by the $^{239+240}$Pu measurements and the inverse modelling. Non-eroded reference profiles (n: 210) in the region show the parent material (calcaric glacial till) at 102 cm on average (van der Meij et al., 2019). Hence, less than 17% of soil depth reduction can be attributed to most recent process dynamics. This suggests that traditional hand or cattle based tillage systems, which are used since the beginning of arable agriculture in the region (1k yr BP; Kappler et al., 2018), must cause severe soil redistribution in the long run. This suggests that tillage erosion might be the dominant process even

without mechanized soil tillage, which is the common practice in most developing countries that also partly cultivate very steep slopes. Therefore, the general assumption of tillage erosion being only an issue for highly mechanised agricultural systems (Van Oost et al., 2006) might need to be re-evaluated.

### 4.3 Interplay of sediment redistribution by water and tillage

The inverse modelling has shown that soil redistribution by water has only a minor impact on erosion processes in the study

area. However, sediment deposition by water has a complex interplay with tillage translocation (kettle hole surrounding flat; Fig. 7). Very high deposition by tillage translocation towards the field-kettle hole edge (typically >1 m known from soil augering; Kappler et al., 2018) builds up local hydrological depressions (Fig. 1 & 7). Only large and rare events exceed the critical runoff quantity to enter the kettle hole, while the majority of events force deposition of sediments in the kettle hole surrounding flats (see Fig. 7d; Fig. 3, P1 and P4). Therefore, these catchments show a very limited hydrological and

sedimentological connectivity between the arable land and kettle holes. Surface runoff and sediment delivery monitoring in the study catchment (2015-2019) have demonstrated that only very few rainfall events produced sufficient runoff to exceed the retention capacity of the kettle hole surrounding flats and caused sediment delivery (data not shown). Hence, tillage translocation in hummocky young morainic regions does also have a pronounced impact on hydrology and biogeochemistry.



### 4.4 Relevance of tillage erosion and scientific attention

Our results clearly indicate that soil erosion in the study area is high and is mainly attributed to tillage erosion (Fig. 6 & 7). During the socialist era (1949-1990), efficient agricultural management strategies were implemented that included land reformation to merge large fields and the use of heavy farming machines (Forstner and Isermeyer, 2000; Wolz, 2013). For instance, annual ploughing was combined with a recommended practice of using a paraplough (tillage depth ~0.6 m; Fachbereichsstandard-DDR, 1985) to break the plough pan. The average field size in the region (Quillow catchment: 22 ha) 350 still consists of large field sizes that has established conventional big farming structures that is based on powerful machinery. However, tillage erosion does not receive reasonable scientific attention (Fiener et al., 2018), even if the effects on yields (Oettl et al., 2020), nutrient and carbon cycling (Wilken et al., 2017a; Zhao et al., 2018; Nie et al., 2019) and soil hydrology (Herbrich et al., 2017) are widely known. Globally, tillage erosion has been recognized as an environmental threat in the wide spread hummocky young morainic regions of Canada: (Pennock, 2003; Tiessen et al., 2007a; Tiessen et al., 2007b), North America 355 (Li et al., 2007, 2008), Russia (Olson et al., 2002; Belyaev et al., 2005) and Northern Europe (Quine et al., 1994; Heckrath et al., 2005; Wysocka-Czubaszek and Czubaszek, 2014) that have shallow soils that are subject to dropping yields at hilltop locations. Most arable regions are subject to pronounced tillage erosion (e.g. illustrated in the landscape by tillage banks along downslope field borders), but may not show a pronounced impact on yields (Lal et al., 2000). For instance, loess derived soils with a homogeneous grain size distribution for several meters of depth do not show major differences in soil structure (Blume 360 et al., 2016), while nutrient losses are compensated by fertilizer applications. Another reason for not being a prominent soil degradation mechanism might be that tillage erosion is simply not as visually present as water erosion, which leads to rapid topographical dynamics (rills and gullys) and off-site damages (muddy floods, siltation). However, tillage erosion is a highly important soil redistribution process, taking place on the majority of sloped arable fields that urgently needs to be (re)drawn on the maps of the scientific community and policy makers.

**5 Conclusions**

Within this study, $^{239+240}$Pu was used as a tracer to determine soil redistribution in a hummocky young morainic study catchment under intense arable use. To understand the role of water and tillage erosion on soil degradation patterns, an inverse modelling approach was carried out in the study catchment. The results clearly show that recent soil degradation in the study area is dominated by tillage translocation. Furthermore, tillage erosion has a substantial impact on surface hydrology as deposition at 370 the edge between the field border and the kettle hole builds up hydrological depressions. These hydrological depressions limit the surface runoff and sedimentological connectivity into the kettle hole and force deposition of sediments transported by water. Soil redistribution by water has no major contribution to soil loss at the slopes of the catchment, but causes pronounced deposition in the spatially narrow area of the kettle hole surrounding flat. Within this study, soil erosion up to 17 cm (53 yr)$^{-1}$ and deposition exceeding 25 cm (53 yr)$^{-1}$ of recent morphodynamics (since 1960s) were found. However, even these relatively 375 high erosion rates cannot explain the current soil degradation patterns that show both profile soil truncation and colluviation



larger than one meter. This indicates that tillage erosion is not a process that exclusively takes place in highly mechanised agro-ecosystems. Our results clearly underline that tillage erosion is a critically underrepresented soil degradation process that takes place on the majority of sloped cultivated land.

## Data availability

The data will be made available on request.

## Author contribution

This paper represents a result of collegial teamwork. FW, PF and MS designed the sampling scheme. FW and SK carried out the field campaign. SK and FW prepared the soil samples for $^{239+240}$Pu activity analysis that were carried out by MK. Data processing and analysis was done by FW. FW and PF prepared the manuscript. All authors read and approved the final
manuscript.

## Competing interest

The authors declare that they have no conflict of interest.

## Acknowledgements

We gratefully acknowledge the support during fieldwork of Norbert Wypler, Lidia Völker and Christoph Kappler. Special
thanks also go to the farm owner Bernd Sohn for his permission to carry out various types of measurements on his field in the Uckermark.



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



**Tables**

Table 1: Parametrisation of the inverse modelling approach.

| Parameter | Standard value or range in inverse modelling | Iteration step | Unit | Source |
|---|---|---|---|---|
| USLE factors | | | | |
| P | 1.0 | | - | Standard value for soil management without specific soil conservation, e.g. contour ploughing |
| C | 0.081 | | - | Calculated from crop rotation following the procedure of Schwertmann et al. (1990) |
| R | 45 | | kJ m$^{-2}$ mm h$^{-1}$ | From erosivity map of BGR (2014a) |
| K | 25 | | kg m$^2$ h m$^{-2}$ MJ$^{-1}$ mm$^{-1}$ | From soil map of BGR (2014b) |
| LS | variable | | - | Calculated using 5 m DEM provided by the state of Brandenburg, Germany |
| Parameters varied during inverse modelling | | | | |
| Factor changing the product of all USLE factors (water erosion strength) | 0.1...2 | 0.1 | - | |
| ktil | 25...1000 | 25 | kg m$^{-1}$ | |
| ktc | 25...500 | 25 | m | |

595



**Figures**

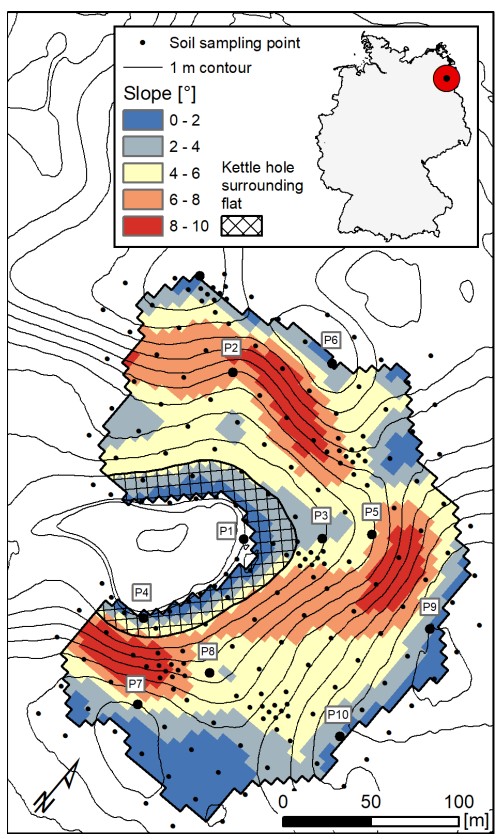

Figure 1: Topography and nested soil sampling scheme in the young morainic study area in northeast Germany. P1

600    to P10 indicate the locations for high-resolution depth profile sampling (see Fig. 3).





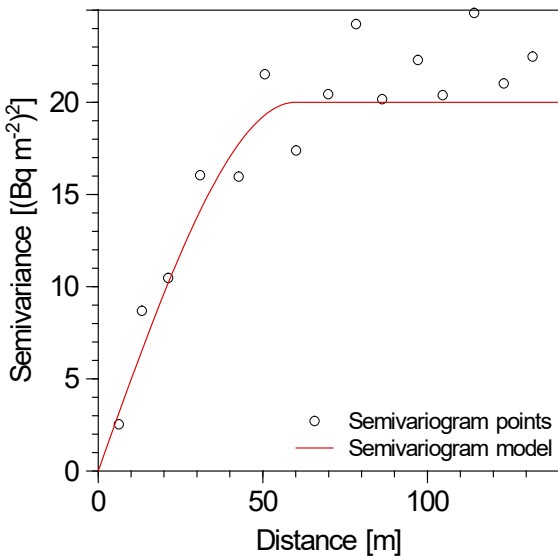

Figure 2: Semivariogram and semivariogram model fit of the [239+240]Pu block kriging interpolation.




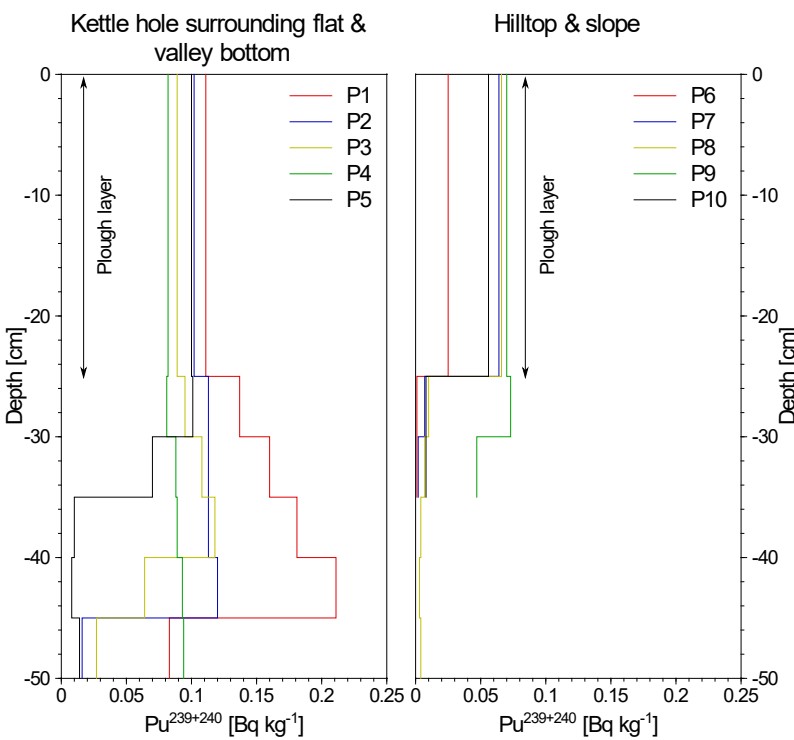

Figure 3: Depth distribution of $^{239+240}$Pu at different geomorphological positions. Locations P1 to P10 are given in Figure 1.





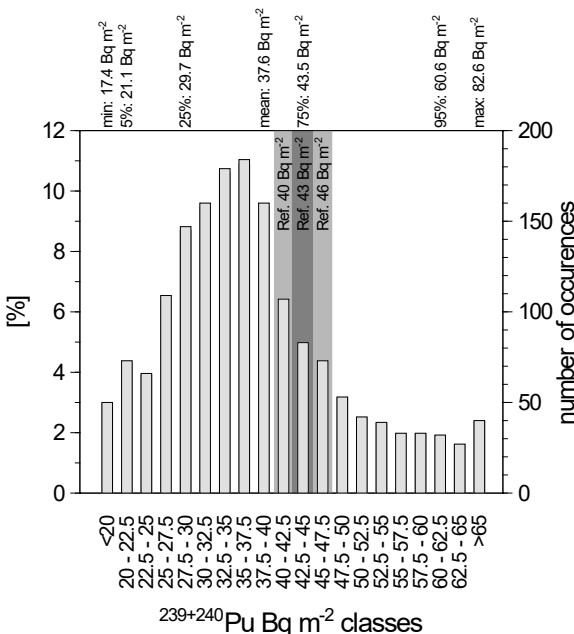

610 Figure 4: Distribution histogram of 5 m x 5 m interpolated $^{239+240}$Pu measurements in 20 classes with descriptive statistics.





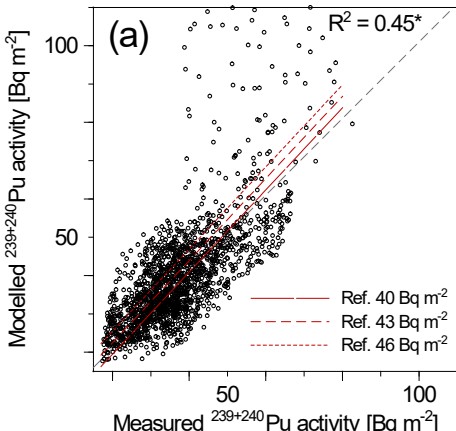

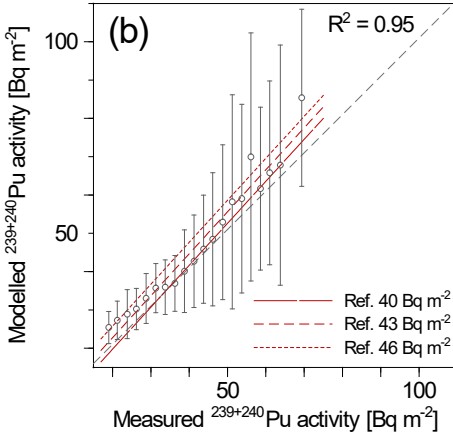

Figure 5: Linear correlation between measured and modelled [239+240]Pu inventories redistributed by water ($ktc$: 150, P-factor: 1) and tillage erosion ($k_{til}$: 350 kg m$^{-1}$; * = p-value < 0.001). (a) Point by point correlation on 5 m x 5 m resolution (n: 1699); (b) class aggregation according to [239+240]Pu derived soil redistribution. Minimum and maximum class n is 27 and 184, respectively. While the points and classes are calculated for a reference of 43 Bq m$^{-2}$, the trend lines display the offset sensitivity of different reference [239+240]Pu activities.





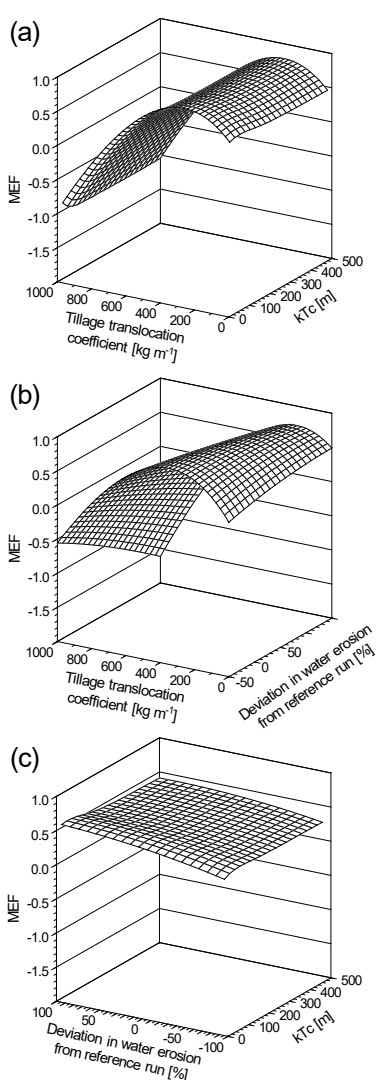

Figure 6: Inverse modelling of tillage and water erosion compared to $^{239+240}$Pu derived soil redistribution. Three parameter combinations (tillage transport coefficient, $k_{til}$; water transport capacity coefficient, $ktc$; deviation in water erosion strength compared to reference parameterisation) are tested on their effect on the goodness of fit, represented by the MEF (model efficiency coefficient: perfect model fit = 1; model prediction as good as the mean = 0; model prediction worse than mean = <0).

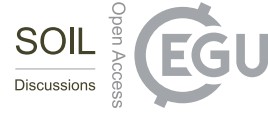

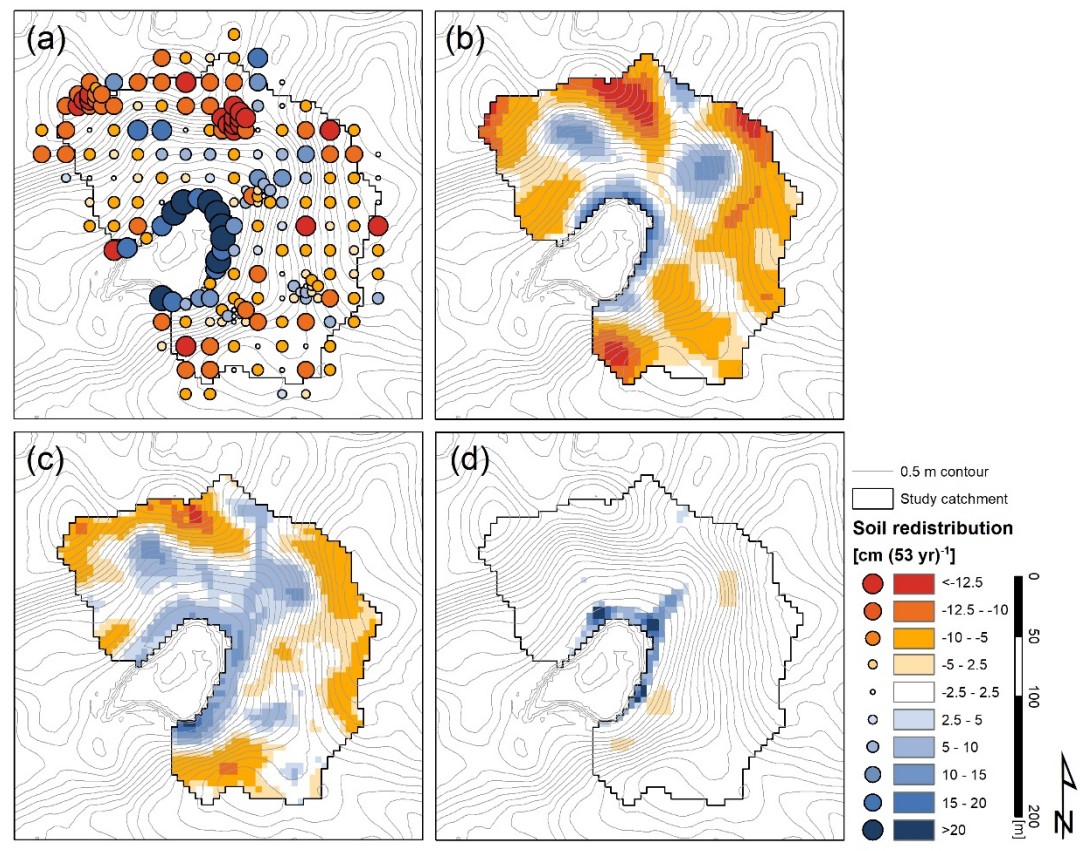

Figure 7: The Figure consists of four parts: (a) Soil redistribution derived from $^{239+240}$Pu top and subsoil measurements using 43 Bq m$^{-2}$ as the reference inventory; (b) geostatistically interpolated soil redistribution based on $^{239+240}$Pu point measurements; (c) modelled tillage erosion with a tillage transport coefficient ($k_{til}$) of 350 kg m$^{-1}$; (d) modelled water erosion according to reference parameterization ($ktc$: 150; also see Table 1).