# Peer review of "Understanding the role of water and tillage erosion from 239+240Pu tracer measurements using inverse modelling"

_SOIL, 2020_

## Referee Comment (RC1) · Anonymous Referee #1 · 7 May 2020

General comments:

I very much enjoyed reading this manuscript. It shows how soil erosion models can be used to understand relevant processes and to test hypotheses about a system – in this case, that soil redistribution in the studied catchment is primarily controlled by tillage erosion. The manuscript is well written in clear fashion and provides a strong contribution to the field. I have a few minor questions and comments, mostly on the modelling side.

Specific comments:

L 82: I don't think parameter space is the term you are looking for here. Consider

changing this to "parsimonious parameter set".

Equation 3: This formulation of WaTEM/SEDEM looks a little different to what I usually see (e.g. Tc = Ktc R K (LS-aSIR)). Is this correct?

Equation 5: Is ktil calculated by this equation? I thought it was being sampled from a pre-defined range during the inverse modelling.

Are the raster outputs from equations 3 and 5 summed? A flow-chart explaining the modelling might be helpful to guide the readers.

L246: Why qualitatively?

L247-253: Are these correlations calculated based on the best-fit model realization? Please inform this in the text.

L256-257: By looking at figure 6c I wouldn't say the predictions showed "hardly any sensitivity to erosion strength". The MEF ranges to -0.2 to 0.6, with higher values clearly associated to the positive deviations in water erosion from reference run. Of course, the effect of the tillage parameter is much more pronounced. Nevertheless, it would be a good idea to show univariate dotty plots of the sampled parameter space with their associated goodness-of-fit measure.

L260: I think it would be nice to look at the parameter space which produced acceptable model realizations (for instance, within the kriging variance or above a given MEF threshold), instead of focusing on the best-fit. That would make your approach more robust. There are multiple solutions to an inverse problem, and perhaps that should be more explicitly recognized. I mean, if you consider the error in the observed data (interpolated Pu map), there might acceptable model realizations produced with the contribution of the water erosion component. This also relates to the discussion in lines 307-310.

L309-310: Not sure I understand this. If none of the sampled parameter values produced adequate system representations, why would we need to assume the reference

parameterisation is the most appropriate?

Technical corrections:

L209: "Results" is repeated.

L307-308: "Inverse modelling" is repetitive at the end of the sentence.
* * *

---

## Referee Comment (RC2) · Olivier Evrard (Referee) · 8 May 2020

General comments This study quantifies soil redistribution due to both water and tillage erosion processes in a 4.2-ha catchment (corresponding to one single field) located in an intensively cultivated region of Northeastern Germany. To reach this goal, Pu-239+240 inventory measurements have been conducted (and interpolated) and an inverse modelling approach (based on the SPEROS-Pu conceptual model) has been undertaken. Overall, the study was well designed (through a nice nested sampling approach with five densified sub-grids). In total, soil cores were collected at 219 locations (including 10 detailed depth profiles sampled at different topographic locations across the study site) and analysed for Pu-239 and Pu-240 by ICP-MS, which represents considerable field and lab work efforts! The authors should be congratulated for that!

At some places, the text is unclear and could be improved, the use of several terms and technical formulations is sometimes misleading, but this will likely be fixed easily by the authors when revising the text (see the annotated pdf file).

In the discussion and in the conclusions, I have the feeling that, at some places, the authors go maybe a bit too far when extrapolating their results, and they should be more nuanced in the text. Importantly, the advantages of using Pu-239+240 inventories (compared to Cs-137 inventories) for reconstructing soil redistribution between 1964-2016 should be better justified in the text, in my opinion. Of note, analysing Pu-239 and Pu-240 requires time-consuming chemical sample preparation steps that are not required for analysing Cs-137 ('simple' physical measurement). A reason for using Pu isotopes could be that the study area received significant Chernobyl fallout in 1986 (in addition to the global fallout with a peak in 1963-64), which would complicate the temporal reconstruction. However, this is not specifically addressed by the authors (nor supported by their measurement of both Pu-239 and Pu-240, the ratio of which should directly provide the answer?) Of course, there could also be other (good) reasons to use Pu isotopes instead of Cs-137, but their clarification in the text would be appreciated.

For detailed comments, questions and suggestions all throughout the text, please refer to the enclosed annotated pdf file.

Please also note the supplement to this comment:
https://www.soil-discuss.net/soil-2020-22/soil-2020-22-RC2-supplement.pdf

**Supplement:**

[revised manuscript text omitted]

---

## Author Response (AR1)

**Point-by-point response Referee #1**

**Understanding the role of water and tillage erosion from $^{239+240}$Pu tracer measurements using inverse modelling**

Florian Wilken, Michael Ketterer, Sylvia Koszinski, Michael Sommer and Peter Fiener

We appreciate that the reviewer acknowledges the study design and supports the publication. The reviewer raised questions with regards to the modelling part of the study. We appreciate the advices and revised the manuscript accordingly. Please see the detailed answers (in italics) to the comments below:

L 82: I don't think parameter space is the term you are looking for here. Consider changing this to "parsimonious parameter set".

*Thanks, we follow the suggestion!*

Equation 3: This formulation of WaTEM/SEDEM looks a little different to what I usually see (e.g. Tc = Ktc R K (LS-aSIR)). Is this correct?

*Our version of SPEROS is not identical to the WaTEM/SEDEM version of van Rompaey et al. 2001. Our SPEROS version does not include a rill coefficient as for instance used in the SPEROS version by Bouchoms et al. 2017. This was done as (i) rill erosion is not a dominant process in the study area and (ii) the rill/interrill ratio is typically used for longer slopes that cannot be found in the study area. For instance, van Rompaey et al. 2001 use observations from Govers and Poesen 1998 that on a 0.06 m m$^{-1}$ slope after a distance of **65 m** the interrill erosion rate equals the rill erosion rate. To avoid confusion, we remove the reference to WaTEM/SEDEM from the text.*

Equation 5: Is ktil calculated by this equation? I thought it was being sampled from a pre-defined range during the inverse modelling.

*Indeed, ktil was not calculated but implemented for a pre-defined range that covers low to extreme tillage erosion rates. The range was defined with respect to literature values and own experiments. As the ktil is one of the three basic parameters of the inverse modelling approach, we would like to provide the formula of the ktil to assist the reader understanding how this*

*parameter is derived. To avoid confusion, we mention that the ktil formula is just shown for illustration: "(as given for illustration in Eq. 5)" in the implementation section.*

Are the raster outputs from equations 3 and 5 summed? A flow-chart explaining the modelling might be helpful to guide the readers. *Yes, the output of the different parameter combinations was summed up and subsequently statistically analysed (e.g. Fig. 5 & 6).*

*We had an extensive discussed among the authors if a flow-chart would improve the readability of the manuscript. The result of this discussion was that we probably would need a quite detailed flow chart to substantially add a value for the reader. As the model was already described in detail in earlier papers, we would prefer keeping the focus on the model results rather the model description. So, we would prefer not adding an additional figure. However, if the editor also thinks that a flow-chart would be helpful, we would follow his advice.*

L246: Why qualitatively?

*Thank you for pointing at this! We changed the text to "quantitatively"*

L247-253: Are these correlations calculated based on the best-fit model realisation? Please inform this in the text.

*These calculations are based on the best model fit realisation of the ktil and the reference for water erosion. We did not explicitly stress this in the text as the correlation results are only affected by changes in the ktc value (ktil and RUSLE product do not influence spatial soil redistribution patterns) that follow the reference parameterisation. To avoid confusion, we reformulate the sentence: "The spatial correlation between the $^{239+240}$Pu derived patterns and the modelled best knowledge soil redistribution, including both water and tillage erosion, is only moderate ($R^2$: 0.45) on a raster by raster comparison (n: 1699, 5 m x 5 m grid points; Fig. 5a)."*

L256-257: By looking at figure 6c I wouldn't say the predictions showed "hardly any sensitivity to erosion strength". The MEF ranges to -0.2 to 0.6, with higher values clearly associated to the positive deviations in water erosion from reference run. Of course, the effect of the tillage parameter is much more pronounced. Nevertheless, it would be a good idea to show univariate dotty plots of the sampled parameter space with their associated goodness-of-fit measure.

*We agree, there is minor sensitivity that, however, does not lead to any indication of an optimal parameterisation. The requested dotty plots underlines that statement (see below), however, we think the 3D plot nicely shows the interplay between the parameterisations. Hence, we would prefer to stay with the 3D Figure.*

[Figure]

L260: I think it would be nice to look at the parameter space which produced acceptable model realizations (for instance, within the kriging variance or above a given MEF threshold), instead of focusing on the best-fit. That would make your approach more robust. There are multiple solutions to an inverse problem, and perhaps that should be more explicitly recognized. I mean, if you consider the error in the observed data (interpolated Pu map), there might acceptable model realizations produced with the contribution of the water erosion component. This also relates to the discussion in lines 307-310.

*In general, we agree that there are multiple solutions for an inverse modelling and this might result in a substantial equifinality problem. However, as we did only vary three parameters where the parameter for the tillage erosion clearly dominates soil redistribution processes, there is no typical equifinality problem resulting from our approach. We follow the idea to provide a range of the ktil parameterisation that achieves a MEF better than 0.8 and demonstrate the limited sensitivity of the water erosion parameterisation on the best-fit model run. The following was added to the text: "A MEF better than 0.8 and RMSE below 6.5 Bq m$^{-2}$ were found for a $k_{til}$ range from 225 to 475 kg m$^{-1}$, while the best model fit was found for a $k_{til}$ of 350 kg m$^{-1}$ achieving a MEF of 0.87 and a corresponding RMSE of 5.2 Bq m$^{-2}$. The best model fit was found without the contribution of water erosion. The highest impact on the best-fit model run was a 0.31 MEF-reduction by an extreme water erosion parameterisation ($k_{tc}$ = 500, water erosion strength = 200%)."*

L309-310: Not sure I understand this. If none of the sampled parameter values produced adequate system representations, why would we need to assume the reference parameterisation is the most appropriate?

*Thanks for pointing at this. As the inverse modelling did not indicate any optimal value for the water erosion parameterisation, we follow the input data recommendations that has been officially published by the administration of the state Brandenburg, Germany. We changed this in the text: "Therefore, we assume the reference parameterisation for the region given by the state of Brandenburg as the most appropriate (ktc: 150 m and RUSLE parameters according to Tab. 1)."*

Technical corrections:

L209: "Results" is repeated.

*Thanks, we changed the text as follows: "[…]raster map against the results calculated by the inverse modelling approach."*

L307-308: "Inverse modelling" is repetitive at the end of the sentence.

*Thanks, we changed the text as follows: "This sediment delivery into the kettle hole cannot be explained by the inverse modelling of water erosion applying a reasonable parameter range."*

**Point-by-point response Referee #2**

**Understanding the role of water and tillage erosion from 239+240Pu tracer measurements using inverse modelling**

Florian Wilken, Michael Ketterer, Sylvia Koszinski, Michael Sommer and Peter Fiener

We highly appreciate the comprehensive and very valuable review carried out by Olivier Evrard, thank you! The reviewer positively highlights the study design and efforts made but has concerns about the conclusions on the contribution of soil degradation before the 1960[th]. We revised the manuscript accordingly. Please see the detailed answers (in italics) to the comments below:

**Abstract:**

L14: Something wrong here; *we rephrased the text as follows, "The majority of soil redistribution studies focus on water erosion,[…]"*

L17: why 'whereas'?; *we changed whereas to "where"*

15 L19: Suggestion, types; *thanks, we follow the suggestion*

L19: Suggestion, individual; *thanks, we follow the suggestion*

L20: Suggestion, soil redistribution patterns; *we would like to keep this phrase. The effect of lateral soil translocations on the (regular) soil distribution (soil pattern) in landscapes is meant here, not the process dynamics/rates.*

20 L23: Suggestion, this; *we changed the text to: "[…]the study catchment[…]"*

L24: Unclear why you refer to this 'kettle hole' for the 1st time here; *we introduce the kettle hole in the study area description: "The aim of this study is to understand the contribution of water and tillage*

*erosion leading to soil patterns found in a small hummocky ground moraine kettle hole catchment under intensive agricultural use."*

L26: I would clarify this sentence to make it more explicit... Do you mean that tillage erosion started before the mechanisation in the 1960s?; *We follow the recommendation and reworded the sentence as follows: "Hence, tillage erosion already started before the onset of intense mechanisation since the 1960s."*

L28: could be more explicit what you mean here; *we know from other studies that tillage erosion is the dominant driver of soil properties (SOC, SIC, N, texture, micro- & macro aggregates) and finally crop yields in the region. However, within this study we focus on erosion patterns derived from FRN distributions that does not explicitly deal with fertility or other soil properties. Hence, we would like to avoid statements that are not covered by our analysis. We reduced the sentence as follows: "In general, the study stresses the urgent need to consider tillage erosion as a major soil degradation process that can be the dominant soil redistribution process in sloped arable landscapes."*

**Introduction:**

L34: I agree with you but I think that this is more general than related with EU WFD... in many countries and regions, soils/agricultural land are managed by different authorities than waterbodies, which complicates the design of integrated approaches at the catchment scale where the connection between cultivated hillslopes and river systems would be taken into account...; *thanks for this comment. We agree that an integrated catchment management is also limited due to administrative barriers. We changed the text as follows: "Within the European Union, the focus on off-site erosion effects is partly caused by the definition of the goals of the EU Water Framework Directive (EU 2000/60/ES) that focuses mainly on water bodies and floodplains but not on a fully integrated catchment management that would call for complex shared responsibilities between different administrative units."*

L36: not very clear; *We reformulate the text as follows: "Thereby, other soil erosion drivers like tillage and wind are somewhat out of the scope of most studies."*

L39: Unclear; *We changed the text as follows: "Particularly areas of a hummocky topography with short summit-footslope distances, such as young morainic areas, […]"*

L41: Suggestion, associated; *thanks, done*

L50: Internal redistribution within the catchment?; *To be more specific, we changed the text as follows: "[…]address catchment internal redistribution."*

L53: Suggested change: "cannot provide a reconstruction of soil redistribution changes throughout time?"; *we changed the text as follows: "However, these beforehand applied tracers and change monitoring methods cannot provide a reconstruction of soil redistribution of the past."*

L53: Unclear what you mean here?; *we agree and removed that part of the sentence: "Natural or anthropogenic tracers in soils can be used to understand soil redistribution."*

L55: Parsons and Foster 2011, not sure this is the most appropriate article to support your statement here!; *thanks, you are right. We remove this reference.*

L56: Could be rephrased here to clarify; *we rephrase the sentence as follows: "The force of atmospheric nuclear weapon tests transported radioisotopes outside the troposphere, where circulation led to a globally almost homogeneous spatial distribution and subsequent fallout on soils by rain (Alewell et al. , 2017)."*

L57: Why 'limited' test ban treaty and suggestion to write it in capitals; *To our knowledge , the Limited or Partial Test Ban Treaty still allowed underground detonations. In 1996 all kind of nuclear weapon tests were banned by the Comprehensive Test Ban Treaty. However, as atmospheric tests and the year is named, we think we can only refer to the Test Ban Treaty to avoid confusion by the word "limited". We changed the text as follows: "[…]while the Test Ban Treaty caused[…]"*

L59: Of note, there was (much less) but still significant fallout after 1963, until the early 1980s. We've recently figured out that a lot of radionuclide atmospheric monitoring data is lacking for documenting this fallout, mainly for the periods between 1954-1959 and 1967-1976 (see Evrard et al., 2000,Fig. 1-2 and associated text; https://doi.org/10.1016/j.geomorph.2020.107103).; *Thanks, we moved from rapid end to "rapid decrease" and refer to the paper.*

L60: […]which enables the use of radioisotopes as redistribution tracers in soils. Reference missing here.; *Thanks, we added Meusburger et al. 2016*

L61: Suggestion to restrict the reference list to three items, (e.g. Van Oost and Govers, 2006; Porto and Walling, 2012; Zhang, 2015; Greenwood and Meusburger, 2019; Srivastava et al., 2019); *we followed the suggestion: (e.g. Porto and Walling, 2012; Chartin et al., 2013; Evrard et al., 2020)*

L66: I disagree with you unless you add 'in those areas that received significant Chernobyl fallout' (is this what you mean?); *the fallout of the Chernobyl disaster shows a high spatial variability but the majority of regions in Europe were to some extent affected. As the unmixing of the 1960$^{th}$ and Chernobyl $^{137}$Cs is complex, we think the use of $^{137}$Cs is in Europe subject to uncertainties. However, it is not our intention to say that the use of $^{137}$Cs in Europe is generally impossible, however, requires special care. To indicate this, we changed the text as follows: "Hence, the use of $^{137}$Cs as a soil redistribution tracer in Europe is associated with uncertainties and requires special attention concerning a potential Chernobyl contamination (Evangeliou et al., 2016)."*

L67: Could you quantify this in %? (if you calculate the decay between 1963 and 2000); *thanks, for this good idea. We added in brackets that about 70% of the global fallout has been reduced.*

L69: Does not make sense to refer to the half-life of a sum of radionuclides, does it?; *thanks for this hint, we changed the text as follows: "[…]the half-life of $^{239}$Pu and$^{240}$Pu is long [...]"*

L70: Could you be more explicit here?; *we provide the radius of pronounced $^{239}$Pu and $^{240}$Pu contamination surrounding Prypiat, Chernobyl: "(<100 km; Kashparov et al., 2004; Matsunaga and Nagao, 2009)"*

L74: Unclear, please rephrase; *we rephrased the text as follows: "However, the use of radioisotope tracers integrates all types of soil redistribution processes and does not provide information on the relative contribution of the driving processes at play (e.g. water, tillage, wind)."*

L75: Suggestion, Unravelling the respective contributions of these different processes […]; *thanks, done*

L77: Unclear; *we rephrased the text as follows: "There are only few models that take both water and tillage erosion processes into account. Physically oriented models like MCST-C (Wilken et al., 2017b)*

*and LandSoil (Ciampalini et al., 2012) simulate individual erosion events and are developed to enhance process understanding, while conceptual USLE based models (WaTEM/SEDEM: Van Oost et al., 2000; Van Rompaey et al., 2001) aim at a robust prediction of long-term soil erosion rates (Alewell et al., 2019)."*

5    L82: Range/variation? is this what you mean?; *we want to point out that a model to unravel the role of water and tillage erosion over more than 50 years should be based on an easy to understand and available parameter set (space). For instance, within this study a single tillage erosion and two water erosion parameters were iterated. We changed the text as follows to avoid confusion: "[...]with a limited parameter space covered by available input data[...]"*

10   *L85: Suggestion, objective; thanks, done*

**Methods:**

L92: Unclear what you mean here, as I would say that this is the case for any catchment; to indicate that the small catchment shows high morphological variability; *we changed the text as follows: "The study*

15   *area is part of a kettle hole catchment (4.2 ha; Fig. 1) showing a high morphological variability covering convex hilltops, steep slopes and flat areas."*

L97: Suggestion, are explained by land consolidation programmes implemented in the 1960s[...]; *Thanks, we followed the suggestion.*

L98: How large were the farming structures?; *We provide the following information in the text: "In 1939,*

20   *large farms that manage more than 100 ha of arable land cultivated 7% of the total arable area of Germany. In the present-day federal states of Mecklenburg-Pomerania, Brandenburg and Saxony-Anhalt, large farms cultivated 30% of corresponding arable land (Wolz 2013)."*

L100: We would almost need a sketch with the location of the different soil types along the catena: *we follow the recommendation and add an idealised catena to the study area description (Fig. 1b).*

25   L108: Suggestion, take place each year?; *thanks, we followed the suggestion*

L111: Do you mean 'per transect line'? maybe add that it is to avoid 'boundary effects'?; *we changed the text as follows: "The soil sampling design was organised according to a regular 20 m x 20 m grid with at least one sampling point of the transect line exceeding the spatial extent of the catchment under study (Fig. 1) to avoid boundary effects."*

L112: This is cool!! *Thanks for appreciation!*

L118: Suggestion, ranging; *thanks, we follow the suggestion*

L122: Sorry if I missed something, but I think that you shouldn't use the term 'replicate' if you collected one single core at each of these 10 different locations, right?; *We took at 209 locations 219 cores. The replicate cores were taken at ten selected sites in direct proximity to each other (max. 50 cm apart) and were supposed to be representative for the same location. This was basically done to achieve more soil material for additional analysis like the high resolution 5 cm increment vertical resolution measurements. The replicate cores are not critically relevant for the analysis. Hence, we decide to reduce the information to avoid confusion: "Closed soil cores, using a steel drill containing a plastic liner (4.6 cm inner diameter), were driven by a percussion corer (Cobra TTe; Atlas Copco Power Techniques GmbH, Stockholm, Sweden) into the ground down to a depth of 50 cm at 209 sampling points."*

L127: Suggestion, n=3; *thanks, we follow the suggestion*

L129: Suggestion, air-dried; *thanks, we follow the suggestion*

L133: Suggestion, measured for their; *thanks, we follow the suggestion*

L133: I guess that you did not want to reduce the inventory of samples, but instead the number of samples for which the analysis results would be under the detection limits, right?; *Yes, sure, thanks for pointing at this error. We reformulated the text as follows: "This was done to reduce the number of samples with Pu activities below the detection limit."*

L134: Suggestion, higher-resolution; *thanks, we follow the suggestion*

L154: Suggestion, equations; *thanks, we follow the suggestion*

L158: What about the undisturbed sites? You did not refer to these sites when described the sampling plan?; *We are using a potential range for the reference concentration based on reference measurements of a study that was carried out nearby. We explain this in the implementation section. To indicate this, we point at the corresponding section in the text: "undisturbed sites in Bq m$^{-2}$ (see implementation section 2.5)"*

L168: SPEROS-C and/or SPEROS-Pu?; *Both SPEROS-C and SPEROS-Pu use the same tillage erosion module. However, as the description is for SPEROS-Pu, we only name this version in the text. Thanks, for pointing at this.*

L175: Suggestion, 10-cm depth layers?; *thanks, we follow the suggestion*

L176: Unclear what you mean here?; *Thanks! The word "same average" might have been caused the confusion. We remove the word "same" to indicate that the plough layers have the average $^{239+240}$Pu activity of the two topsoil layers: "[…]have the average 239+240Pu activity of the upper[…]"*

L179: Something missing here; *thanks, we changes the text as follows: "Soil erosion processes lead to a reduction of the $^{239+240}$Pu inventory per m$^2$ due to soil and associated $^{239+240}$Pu loss, which causes mixing in of non-contaminated subsoil."*

L183: Suggestion, topsoil?; *thanks, we follow the suggestion*

L183: Suggestion, was?; *thanks, we follow the suggestion*

L185: Ok, based on how many measurements; *Calitri et al. 2019 determined the reference value based on four sites that showed no geochemical and soil profile morphological (e.g. soil profile truncation) indications for erosion or deposition. We changed the text accordingly: "The reference inventory of undisturbed sites follows the value determined by Calitri et al. (2919) who found a $^{239+240}$Pu inventory of 43±3 Bq m² based on four sites that did not show profile morphological or geochemical indication for soil redistribution at a location 8.5 km apart from the study area."*

L186: Please rephrase; *we changed the text as follows: "To address the uncertainty inherent to the reference measurements, a reference range from 40 to 46 Bq m² was accounted for in the simulations."*

L191: Suggestion, a decametre scale?; *thanks, we follow the suggestion*

L191: Something wrong here in the sentence? & missing word here?; *We rephrase the sentence as follows: "Different block sizes were tested for the kriging approach. A block size of 20 m was selected that matches the sampling resolution and did not to cause over-smoothening of the interpolation result."*

L208: Suggestion, goodness-of-fit?; *thanks, we follow the suggestion*

L217: Please provide the manuscript section number instead; *thanks, we follow the suggestion*

**Results:**

L222: Important, do they correspond to those points having the lowest Pu inventories in topsoil layers?; *Yes, the subsoil locations that fall below the detection limit show low inventories below 40 Bq m². Please note that we excluded 30% of the subsoil samples that showed the lowest topsoil activities. Hence, for locations of the lowest inventories, no subsoil measurements were carried out. However, to indicate this relationship we add the following sentence: "Those seven samples are all located at positions with $^{239+240}Pu$ inventories below the lower reference boundary (40 Bq m$^{-2}$)."*

L225: unclear why you use 'in turn' here?; *we changed the text as follows: "All high-resolution depth profiles (5 cm increments) at erosional sites show a sharp reduction of the $^{239+240}Pu$ activity below the plough layer (Fig. 3), while depositional sites show more complex depth distributions."*

L227: Suggestion, reflect; *we changed the text as follows: "[…]with depth that is potentially caused by Pu enrichment processes[…]"*

L232: Suggestion, water and tillage soil redistribution processes?; *thanks, we follow the suggestion*

L232: Sorry; maybe again I miss something here, but isn't there a contradiction between the first and the second sentences here?; *we fully agree that these two sentences are confusing. We simplified the text as follows: "At 14 (of 209) sampling locations, a higher subsoil than topsoil $^{239+240}Pu$ activity was found, which points at deposition of $^{239+240}Pu$ depleted sediments. The majority of these locations show enriched $^{239+240}Pu$ activities in the subsoil (11 of 14).*

L236: Suggestion, induce?; *we changed the word to "cause"*

L239: Unclear what you mean here... Furthermore, I had the impression that you assimilated the tillage depth to 20 cm (2 layers of 10 cm depth) in the above-mentioned text?; *we determined the tillage depth at each sampling location, which was found at an average depth of 23.5 cm. As the model takes 10 cm layers into account, the tillage depth was set to 20 cm in the implementation. We think the information is not critically important why we removed it to avoid confusion.*

L239: Isn't it a bit short here? Why would the enrichment be worse at these 5 locations compared to others? Including those highly enriched locations exceed the methodological maximum of detectable deposition; *As these locations are clearly subject to enrichment processes that cannot be adequately addressed, we think it is better to remove them and interpolate them from the surrounding information. Furthermore, these points were exclusively located near the kettle-hole border which does only exclude small areas. To make this clearer, we changed the text accordingly: "As the enrichment processes inherent to these five locations cannot be corrected, the locations were excluded from the analysis."*

L247: Suggestion, soil redistribution; *thanks, we follow the suggestion*

L248: Seems to be part of the discussion?; *Our motivation to introduce this before the discussion section was to support the reader to understand the step of analysing classes. However, from a manuscript structural perspective this is indeed not correct, which is why we remove the sentence.*

L252: Suggestion, illustrates the good agreement...?; *thanks, we follow the suggestion*

L255: Description of the MEF in the methods; *we think it helps the reader to see the description of the MEF in direct proximity to the numbers. Furthermore, the MEF is rather a standard parameter that might be familiar to the majority of readers. Hence, we would prefer to keep the short description in the results section.*

L261: could you clarify what you mean here?; *we changed the text throughout as follows: "[...]geomorphological dynamics[...]"*

L262: Suggestion, reach; *thanks, we follow the suggestion*

L263: ?; *we changed the text as follows: "Soil erosion at hilltop locations is shown to reach up to 14.9 cm (43 Bq m$^{-2}$ reference; 40 Bq m$^{-2}$ reference: 14.1 cm; 46 Bq m$^{-2}$ reference: 15.6 cm), while deposition can build a colluvium layer with a maximum thickness of 21.5 cm (43 Bq m$^{-2}$ reference; 40 Bq m$^{-2}$ reference: 24.9 cm; 46 Bq m$^{-2}$ reference: 18.6 cm) over the past 53 yr."*

**Discussions:**

L271: this remains questionable in my opinion (see my general comment), general reviewer comment: Importantly, the advantages of using Pu-239+240 inventories (compared to Cs-137 inventories) for reconstructing soil redistribution between 1964-2016 should be better justified in the text, in my opinion.

10   Of note, analysing Pu-239 and Pu-240 requires time-consuming chemical sample preparation steps that are not required for analysing Cs-137 ('simple' physical measurement). A reason for using Pu isotopes could be that the study area received significant Chernobyl fallout in 1986 (in addition to the global fallout with a peak in 1963-64), which would complicate the temporal reconstruction. However, this is not specifically addressed by the authors (nor supported by their measurement of both Pu-239 and Pu-240,

15   the ratio of which should directly provide the answer?) Of course, there could also be other (good) reasons to use Pu isotopes instead of Cs-137, but their clarification in the text would be appreciated.; *Thanks for the comment and suggested discussion points. We include a section at the beginning of the discussions:*

*"**4.1 $^{239+240}$Pu methodological benefits and limitations***

*The use of fallout radionuclides to determine soil redistribution patterns and rates over the past decades*

20   *has been used in many studies in various study areas around the world (see reviews: Mabit et al., 2014; Alewell et al., 2017; Evrard et al., 2020) and contributed substantially to understand soil degradation processes. However, the most frequently used fallout radionuclide $^{137}$Cs faces upcoming limitations (Chernobyl fallout that adds on the global fallout over large parts of Europe and ongoing decay below detection limit of standard measuring devices; also see section 1) in the use as a soil redistribution tracer*

25   *(Evrard et al., 2020). The fallout radionuclide $^{239+240}$Pu has demonstrated its suitability to determine the recent soil redistribution history (since the 1960s; see review Alewell et al., 2017 and is a potential*

*alternative for $^{137}$Cs as a soil redistribution tracer (Mabit et al., 2013; Alewell et al., 2017). In Europe, where large parts were re-contaminated by $^{137}$Cs fallout of the Chernobyl accident (Evangeliou et al., 2016), additional information on the spatial change on the inventory is needed to derive accurate soil redistribution rates. Particularly in the area of the former GDR, almost no information that can be used*

5 *for a correction on the $^{137}$Cs Chernobyl re-contamination are available (Evangeliou et al., 2016). The $^{239+240}$Pu fallout caused by the Chernobyl disaster was very local (approximate radius of 100 km) and has a distinct fingerprint based on the 239Pu/240Pu ratio. While the 240Pu/239Pu ratio of global fallout in the Northern Hemisphere is 0.180±0.014 (Kelley et al., 1999), the 240Pu/239Pu ratio soils that received high Chernobyl fallout is about twice as high (0.408±0.003, determined for soils within the 30*

10 *km exclusion zone of the Chernobyl reactor; Muramatsu et al., 2000; Boulyga and Becker, 2002). The 95% interval of confidence and average of the $^{240}$Pu/$^{239}$Pu ratio found in the soil samples of this study were 0.281 and 0.199, respectively. Hence, a relevant $^{239+240}$Pu re-contamination by Chernobyl fallout can be ruled out for the study area. Another limitation for the use of $^{137}$Cs as a soil redistribution tracer is the ongoing decay due to short half-life times that has already caused a substantial reduction of the*

15 *inventory. Due to lower activities, measuring devices of much higher complexity are needed in the future (Evrard et al., 2020). Decay is not an issue for $^{239}$Pu and $^{240}$Pu as both nuclides have long half-life times that allow for a quasi-unlimited use, however, it needs to be mentioned that sample preparation for $^{239+240}$Pu ICP-MS measurements is much more laborious compared to the standard procedure of physical measurement $^{137}$Cs measurements.*"

20 L276: OK, I agree with you, but in the real landscapes, maybe the situation is a bit more nuanced, with the redistribution of organo-mineral complexes across landscapes?, *we fully agree and added the following information to the text: "However, it needs to be mentioned that radionuclide associated particles are typically not transported as primary particles but in soil aggregate complexes (Hu and Kuhn, 2014; Hu et al., 2016), which has a pronounced effect on enrichment processes (Wilken et al.,*

25 *2017b)."*

L281: Suggestion, = enrichment in fine particles containing Pu-239+240?; *we slightly modified the suggestion: "enrichment in fine particles of relatively high $^{239+240}$Pu activity"*

L286: OK but is this reasonable if the max erosion depth is 14.9 cm (previous section) and that radionuclide have been homogenized in the entire tilled layer (~25 cm)?; *Yes, we think this is very likely as these highly eroded sites have a reduction of their inventory down to 12 Bq m$^{-2}$, which means 28% of their reference inventory (43 Bq m$^{-2}$).*

L291: Suggestion, flow across?; *thanks done*

L294: Suggestion, was exceeded at 4 sampling locations?; *thanks done*

L299: Suggestion, managed; *thanks done*

L302: Suggestion, both water and tillage erosion processes; *thanks done*

L303: Suggestion, zone/area; *we changed the text to: "zone"*

L304: Could you define this concept at some point in the text?; *We changed the wording to: "geomorphological dynamics"*

L304: Do you mean the limited spatial extent where this process takes place?, *Thanks, we follow the suggestion: "As a result of the small spatial extent where this process takes place, [...]"*

L306: Suggestion; an active process?, *we changed the text to "important"*

L308: Suggestion, in?; *thanks, done*

L310: Doesn't it seem a bit low compared to the erosion depths that you estimated at hilltop locations? Which 'peat' (in terms of spatial extent) are you referring to here?; *We refer to the inner kettle hole that is not under arable use. Hence, 1.7 cm (53yr)$^{-1}$ of colluvium is just based on sediments that were exported from the arable land, therefore, solely transported by water. To make this clearer we changed the text as follows: "According to the model run using the reference parameterisation for water erosion, a colluvial layer of 1.7 cm (53 yr)$^{-1}$ would have been developed on top of the peat that has been exported from the arable part of the catchment (see Fig. 1) due to water transport over the past decades. Furthermore, we highlight the inner peat area of the kettle hole in the study area Figure 1.*

L313: In turn, ?; *we changed the text to "In contrast, "*

L314: Suggestion, have been; *thanks, done*

L320: Suggestion, cover; *thanks, done*

L323: Suggestion, is partially mixed; *thanks, done*

L325: Suggestion, the most eroded sites; *thanks done*

L326: Suggestion, is found at 102 cm depth on average?; *thanks, done*

L329: Suggestion, have caused extensive soil redistribution over long periods?; *thanks, done*

L332: Suggestion, reconsidered across a range of contrasted agricultural environments?; *thanks, done*

L337: Suggestion, unfrequent extreme events; *thanks, we followed the suggestion but used the word "infrequent" instead*

L338: Do you mean to connect hillslopes with the kettle hole?; *we change the text as follows: "Only infrequent extreme events exceed the critical runoff quantity to connect the arable hillslopes with the inner peat area of the kettle hole (Fig. 1),[…]"*

L338: Suggestion, lead to; *thanks, done*

L339: Why using plural here (you investigated one catchment?); *thanks, you are right. We changed the text as follows: "Therefore, the study catchment shows a very limited hydrological and sedimentological connectivity between the cultivated area and kettle hole."*

L340: This was already written earlier in the text (although slightly differently), maybe avoid repetitions in the text at this stage; *we restructured the text and removed the redundant information as follows: "This statement is supported by surface runoff and sediment delivery monitoring in the study catchment (2015-2019) that has demonstrated that only very few rainfall events caused runoff and associated sediment delivery in the kettle hole (data not shown). Therefore, the study catchment shows a very limited hydrological and sedimentological connectivity between the cultivated area and kettle hole."*

L343: could you be more specific here?; *we intentionally kept this statement loose to remain the interpretation of potential processes to the reader as we did not specifically investigate on these effects. Therefore, we would prefer to keep the statement as is but provide the reader additional literature:*

L344: Suggestion, impact? Extent? Magnitude?; *thanks, we changed the section title to "4.4 Impact of tillage erosion and scientific attention"*

L345: Do you mean above the tolerable soil loss rates?; *thanks, we reformulated the text as follows: "soil erosion in the study area exceeds the tolerable soil loss rates (according to Schwertmann et al., 1990: 6 Mg ha-1 yr-1 in the study region) and is mainly attributed to tillage erosion (Fig. 6 & 7)."*

L346: Suggestion, efficient or productivist?; *thanks, we changed the text to "productivist"*

L347: Suggestion, consolidation; *thanks, done*

L350: Suggestion, is rather large?; *thanks, done*

L350: Maybe rephrase here to facilitate reading?; *we reformulated the text as follows: "The average field size in the region (Quillow catchment: 22 ha) is rather large that has favoured big farming structures that utilises powerful machinery."*

L351: Suggestion; likely does not receive...?, *we would like to keep the statement more a bit more drastic to underline the need and prefer to keep the wording as is.*

L351: Suggestion, its; *thanks, done*

L353: Wide spread?, *we removed the word "wide spread"*

L356: Maybe rephrase to clarify here; *we restructured the text to improve the reading flow: "Globally, tillage erosion has been recognized as an environmental threat in the hummocky young morainic regions that have shallow soils that are subject to dropping yields at hilltop locations (Canada: Pennock, 2003; Tiessen et al., 2007a; Tiessen et al., 2007b, North America: Li et al., 2007, 2008, Russia: Olson et al., 2002; Belyaev et al., 2005 and Northern Europe: Quine et al., 1994; Heckrath et al., 2005; Wysocka-Czubaszek and Czubaszek, 2014)."*

L358: Of note, this issue was investigated using radionuclide inventories and the SPEROS model in agricultural regions of central France (Chartin et al., 2013; 10.1016/j.catena.2013.06.006). Of note, these authors also demonstrated the dominance of tillage erosion processes (>90% of the total erosion) in these agricultural landscapes...; *thanks, for naming this very interesting paper! We included the reference here and mention the results of the study in the introduction.*

L358: Suggestion, indeed*?; thanks, we removed the filler*

L361: Suggestion, the impacts of tillage erosion may not be as visible as those caused by water erosion?; *thanks, done*

L363: Unclear what you mean here?; *we reformulated the text as follows: "[...]needs scientific consideration and implementation in soil conservation management by policy makers."*

L370: I didn't understand this when reading the previous text... do you mean that there would be some kind of 'sapping' process along the borders of the kettle hole? If so, this should be clarified in the text (here and above) and better underlined when describing Fig.7c/d...; *no there is no kind of 'sapping' process. Tillage forms a geomorphological flat area, what we call the 'kettle hole surrounding flat'. Within this area, hydrological sinks are formed by tillage translocation that provide water retention capacity, which leads to a reduction of the flow velocity and therefore deposition of sediments that were transported by water. To clarify this, we reformulated the text as follows: "Furthermore, tillage erosion has a substantial impact on surface runoff. Tillage forms hydrological depressions at the downslope border between the cultivated field and the kettle hole that limits the hydrological and sedimentological connectivity into the kettle hole and causes deposition of sediments that are transported by water."*

L372: Suggestion, on the catchment hillslopes?; *thanks, done*

L375: Maybe you go too far in terms of extrapolation based on your results here? More than 1 m, I don't remember to have seen that above (you show a max of 17% of the profile truncation, if I understood it well?); *yes, the maximum erosion that was determined by using $^{239+240}$Pu is 0.17 m. However, extreme erosion and deposition of more than 1 m has taken place in the study area, which was detected based on lab analysis (e.g. $CaCO_3$) and soil prospection that clearly show buried fossil topsoil layers deeper than*

*1 m. Our main argument is that we detected high erosion rates over the past decades (up to 0.17 m), however, those rates can still not explain the current soil degradation stage (>1 m soil profile truncation and burial). To make it clearer that 1 m of erosion or deposition was not determined using FRNs but soil prospection and lab analysis, we clarify this in the text as follows: "[…]soil degradation patterns*

5   *determined from soil prospection and chemical analysis that show[…]"*

L376: Still, your study area can be described as a highly mechanised agro-ecosystem, right? Maybe you go again a bit too far here?; *we see the point but are still convinced that tillage erosion is not just an issue for mechanised agro-ecosystems. Our results indicate an onset of tillage erosion before the main period of agricultural mechanisation (somewhere around the 1960th). We try to rephrase the statement more*

10   *nuanced: "This indicates that tillage erosion might not be a process that exclusively takes place in highly mechanised agro-ecosystems but is potentially causing pronounced soil degradation in smallholder farming structures."*

L378: Again, I would be more nuanced here as well...; *we agree and changed the text accordingly: "Our results clearly underline that tillage erosion is a critically underrepresented soil degradation process that*

15   *can be the main soil redistribution driver on catchment scale."*

[revised manuscript text omitted]

Measuring or monitoring water and tillage erosion is challenging as both processes are interlinked and are strongly controlled by topography (Van Oost et al., 2005b; Van Oost et al., 2006). The quantification of water erosion requires a sufficiently long monitoring time (typically decades) to cover a statistically representative variation of rainfall events occurring on different land cover conditions (Fiener et al., 2019). Therefore, thousands of plot experiments, either driven by natural or artificial rainfall simulations, were carried out in different environments and different land cover conditions (Cerdan et al., 2010; Auerswald et al., 2014). Furthermore, a large number of small catchment studies were performed to  quantify both erosion and depositional processes (for overview see; Fiener et al., 2019). However, soil erosion monitoring is mostly based on sediment delivery  monitoring, which cannot address catchment internal redistribution. In contrast, tillage erosion can only be measured based on the movement of beforehand applied tracers or by morphological change monitoring (for an overview of tillage erosion measuring techniques see; Fiener et al., 2018). However, these beforehand applied tracers and change monitoring methods cannot provide a reconstruction of soil redistribution of the past. Natural or anthropogenic tracers in soils  can be used to understand  soil redistribution (Fiener et al., 2018). Especially anthropogenic radionuclides (e.g. $^{137}$Cs, $^{239+240}$Pu, $^{210}$Pb, $^{7}$Be) have demonstrated their ability to determine changes in topography (Mabit et al., 2014; Alewell et al., 2017; Deumlich et al., 2017). The force of atmospheric Nuclear weapon tests  transported radioisotopes outside the troposphere, where circulation  led to a  (regionally) homogeneous spatial distribution and subsequent  fallout on soils via precipitation (Meusburger et al., 2016). The main period of atmospheric nuclear weapon tests is from 1953 to

1964 (Schimmack et al., 2001), while the test ban treaty caused a rapid decrease of atmospheric bomb tests in 1963-1964 (Wallbrink and Murray, 1993; Evrard et al., 2020). This rapid decrease leads to a distinct peak in the activity of radioisotopes in soils, which enables the use of radioisotopes as redistribution tracers in soils (Alewell et al., 2017). The radioisotope $^{137}$Cs has been used as a soil redistribution tracer in a large number of studies (e.g. Porto and Walling, 2012;

70 Chartin et al., 2013; Evrard et al., 2020) and has become a widely used method in soil erosion science. However, the Chernobyl disaster in 1986 supplied additional radioactive fallout to soils across large areas of Europe (Evangeliou et al., 2016). For some years after the Chernobyl disaster, an unmixing of the Chernobyl fallout from the original 1960s bomb peak signal was possible by the use of the $^{134}$Cs/$^{137}$Cs ratio (Lust and Realo, 2012). However, due to the short $^{134}$Cs half-life of 2 yr (Schimmack et al., 2001), this method cannot be applied anymore. Hence, the

75 use of $^{137}$Cs as a soil redistribution tracer in Europe is associated with uncertainties and requires special attention concerning a potential Chernobyl contamination (Evangeliou et al., 2016). Furthermore, due to the $^{137}$Cs half-life of about 30 yr, decay has already been led to a pronounced reduction (73% in 2020) of the activity until today (Alewell et al., 2017). Over the past decade, $^{239+240}$Pu has been discussed and tested as an alternative radioisotopic tracer for soil erosion studies. Decay is not an issue as the half-life of $^{239}$Pu and $^{+240}$Pu is long ($^{239}$Pu = 24000 yr; $^{240}$Pu = 6563 yr) and the $^{239+240}$Pu contamination by

80 the Chernobyl accident was spatially very limited (<100 km; Kashparov et al., 2004; Matsunaga and Nagao, 2009) and can be determined by the $^{239}$Pu/$^{240}$Pu ratio (Alewell et al., 2014; Alewell et al., 2017).

Radionuclide tracers integrate soil erosion processes over time (e.g. since the bomb peak of the 1960s in case of $^{137}$Cs and $^{239+240}$Pu), which somewhat averages out the large temporal variability of water (episodic nature) and tillage (mechanisation) erosion. However, the use of radioisotope tracers integrates all types of soil redistribution processes and

85 does not provide information on the relative contribution of the driving processes at play (e.g. water, tillage, wind). Unravelling the respective contributions of these different processes requires the use of an inverse modelling approach carrying out model runs with different parameterisations to alter the contribution and mechanisms of different soil redistribution drivers. There are only few models that take both water and tillage erosion processes into account. Physically oriented models like MCST-C (Wilken et al., 2017b) and LandSoil (Ciampalini et al., 2012)

90  simulate individual erosion events and are developed to enhance process understanding, while conceptual USLE based models (WaTEM/SEDEM: Van Oost et al., 2000; Van Rompaey et al., 2001) aim at a robust prediction of long-term soil erosion rates (Alewell et al., 2019). For an inverse modelling approach to unravel tillage and water erosion based on radionuclide tracer, it is necessary to use a parsimonious approach with a limited parameter space covered by available input data over five to six decades, which suggests the use of conceptual models.

95 In this study, we will determine the soil redistribution patterns in a small 4.2 ha catchment based on high-resolution $^{239+240}$Pu measurements and analyse the contribution of water and tillage erosion processes based on an inverse modelling approach using a combined water and tillage erosion model. The general objective is to unravel the importance of water and tillage erosion driving the current variability of soil properties in an intensively used arable landscape of north-eastern Germany.

**2. Methods**

**2.1 Study area**

The study area (53°21'2 N, 13°39'5 E) is situated in the hummocky ground morainic landscape of the Weichselian glacial belt ('young morainic area') of north-eastern Germany (Fig. 1). Characteristic for these landscapes are widespread closed depressions, so-called kettle holes, which result from a delayed melting of dead ice blocks. They are nowadays filled with mineral soil, (degraded) peat or water. The study area is part of a kettle hole catchment (4.2 ha; Fig. 1)  showing a high morphological variability covering convex hilltops, steep slopes and flat areas . The recent crop rotation is rape (*Brassica napus* L.)-winter wheat (*Triticum aestivum* L.)-winter barley (*Hordeum vulgare* L.)-winter barley, cultivated without cover crops, which is a typical conventional crop rotation that is adapted for the highly fertile soils of the Uckermark region. The mean arable land of a farm in the region is 352 ha, which is much larger compared to the mean of the State of Brandenburg (250 ha) and Germany (60 ha; Troegel and Schulz, 2018). These larger field sizes are explained by  land consolidation programmes implemented in the 1960s during the socialist period. However, also before, agriculture in the region was already characterised by large scale farming structures  and corresponding high agricultural mechanisation. In 1939, large farms that manage more than 100 ha of arable land cultivated 7% of the total arable area of Germany. In contrast, within the present-day federal states of Mecklenburg-Pomerania, Brandenburg (study area location) and Saxony-Anhalt, large farms cultivated 30% of corresponding arable land (Wolz, 2013). The catchment is part of a single large field (54 ha), which is a size that can be frequently found in the region. The soils are developed from glacial till and vary with respect to their location in the landscape. Convex hilltops and steep slopes are dominated by extremely eroded A-C profiles (Calcaric Regosols, soil classification according to: IUSS, 2015), while Luvisols showing different degrees of erosion  that are typically situated at the up and mid slopes, the footslopes and depressions  are dominated by Gleyic-Colluvic Regosols (Fig. 1b; Sommer et al., 2008; Gerke et al., 2010). These soils regularly reveal fossil surface horizons below 1 m depth (fAh, fH). 
[revised manuscript text omitted]
$^2$ based on four sites that did not show profile morphological or geochemical indication for soil redistribution at a location 8.5 km apart from the study area. To  address  the uncertainty inherent to the reference measurements , a reference range from 40 to 46 Bq m$^2$ was accounted for in the simulations.

210 The point data of the $^{239+240}$Pu inventory for the depth 0-50 cm was geostatistically (block kriging) interpolated for a gridded spatial representation that matches the spatial resolution of the soil redistribution model (5 m x 5 m). Block kriging was used to reduce small-scale scattering that is naturally inherent to soil cores of 4.6 cm diameter that are supposed to represent a decametre scale. Different block sizes were tested for  the kriging approach. A block size of  20 m was selected that matches the sampling resolution and  did not to  cause  over-smoothening of the interpolation result. The

[revised manuscript text omitted]

To understand the drivers of current $^{239+240}$Pu and associated soil degradation patterns, an inverse modelling was carried out that was  quantitatively analysed by goodness-of-fit parameters. The spatial correlation between the $^{239+240}$Pu derived patterns and the modelled best knowledge soil redistribution, including both water and tillage erosion, is only moderate (R$^2$  = 0.45, Rho = 0.73) on a raster by raster comparison (n  = 1699, 5 m x 5 m grid points; Fig. 5a).  To reduce  small-scale variability and understand the average goodness-of-fit, the inverse modelling results were classified according to the measured $^{239+240}$Pu activity. The classified results average out the spatio-temporal dynamics and show a very high correlation (R$^2$  = 0.95, Rho   0.99; Fig. 5b), which  illustrates the great agreement of the spatial soil redistribution patterns between the $^{239+240}$Pu measurements and the model results.

While the analysis of the spatial correlation is a relative comparison, the absolute deviation is considered according to the MEF (model efficiency coefficient; Nash and Sutcliffe, 1970; 1 = perfect prediction, 0 = as good as mean of all measurements, < 0 = worse than mean). The quality of model predictions shows hardly any sensitivity to water erosion related parameterisations ($k_{tc}$ and erosion strength; Figure 6c). In contrast, the tillage erosion strength, represented by $k_{til}$

parameter iterations, showed a substantial impact on the MEF (Fig. 6a & b). A MEF better than 0.8 and RMSE below 6.5 Bq m$^{-2}$ were found for a $k_{til}$ range from 225 to 475 kg m$^{-1}$, while the best model fit was found for a $k_{til}$ of 350 kg m$^{-1}$ achieving a MEF of 0.87 and a corresponding RMSE of 5.2 Bq m$^{-2}$. The best model fit was found without the contribution of water erosion. The highest impact on the best-fit model run was a 0.31 MEF-reduction by an extreme water erosion parameterisation ($k_{tc}$ = 500, water erosion strength = 200%).

Soil redistribution determined by the proportional conversion approach using $^{239+240}$Pu measurements, indicates substantial  geomorphological dynamics in the study catchment over the past decades. Soil erosion at hilltop locations is shown to  reach up to 14.9 cm (43 Bq m$^{-2}$ reference; 40 Bq m$^{-2}$ reference = 14.1 cm; 46 Bq m$^{-2}$ reference = 15.6 cm), while deposition  can builds a colluvium layer  with a maximum thickness of 21.5 cm (43 Bq m$^{-2}$ reference; 40 Bq m$^{-2}$ reference = 24.9 cm; 46 Bq m$^{-2}$ reference = 18.6 cm)  over the past 53 yr. The inverse modelling stresses that substantial soil erosion, which takes place over large areas, is almost exclusively attributed to tillage translocation (modelled max. water erosion = 3.8 cm (53 yr)$^{-1}$ vs. max. tillage erosion = 13.5 cm (53 yr)$^{-1}$; Fig. 7c, d). In turn, both processes contribute to deposition in the kettle hole surrounding flats (max. water deposition = 27.1 cm (53 yr)$^{-1}$ vs. max. tillage deposition = 22.4 cm (53 yr)$^{-1}$; Fig. 7c, d).

**4 Discussions**

**4.1 $^{239+240}$Pu methodological benefits and limitations**

The use of fallout radionuclides to determine soil redistribution patterns and rates over the past decades has been used in many studies in various study areas around the world (see reviews: Mabit et al., 2014; Alewell et al., 2017; Evrard et al., 2020) and contributed substantially to understand soil degradation processes. However, the most frequently used fallout radionuclide $^{137}$Cs faces upcoming limitations (Chernobyl fallout that adds on the global fallout over large parts of Europe and ongoing decay below detection limit of standard measuring devices; also see section 1) in the use as a soil redistribution tracer (Evrard et al., 2020). The fallout radionuclide $^{239+240}$Pu has demonstrated its suitability to determine the recent soil redistribution history (since the 1960s; see review Alewell et al., 2017) and is a potential alternative for  $^{137}$Cs as a soil redistribution tracer (Mabit et al., 2013; Alewell et al., 2017). In Europe, where large parts were re-contaminated by $^{137}$Cs fallout of the Chernobyl accident (Evangeliou et al., 2016), additional information on the spatial change on the inventory is needed to derive accurate soil redistribution rates. Particularly in the area of the former GDR, almost no information that can be used for a correction on the $^{137}$Cs Chernobyl re-contamination are available (Evangeliou et al., 2016). The $^{239+240}$Pu fallout caused by the Chernobyl disaster was very local (radius <100 km) and has a distinct fingerprint based on the $^{239}$Pu/$^{240}$Pu ratio. While the $^{240}$Pu/$^{239}$Pu ratio of global fallout in the Northern Hemisphere is 0.180±0.014 (Kelley et al., 1999), the $^{240}$Pu/$^{239}$Pu ratio soils that received high Chernobyl fallout is about twice as high (0.408±0.003, determined for soils within the 30 km exclusion zone of the Chernobyl reactor; Muramatsu et al., 2000; Boulyga and Becker, 2002). The 95% interval of confidence and average of the $^{240}$Pu/$^{239}$Pu ratio found in the soil samples of this study were 0.281 and 0.199,

respectively. Hence, a relevant $^{239+240}$Pu re-contamination by Chernobyl fallout can be ruled out for the study area. Another limitation for the use of $^{137}$Cs as a soil redistribution tracer is the ongoing decay due to short half-life times that has already caused a substantial reduction of the inventory. Due to lower activities, measuring devices of much higher complexity are needed in the future (Evrard et al., 2020). Decay is not an issue for $^{239}$Pu and $^{240}$Pu as both nuclides have long half-life times that allow for a quasi-unlimited use, however, it needs to be mentioned that sample preparation for $^{239+240}$Pu ICP-MS measurements is much more laborious compared to the standard procedure of physical measurement $^{137}$Cs measurements.

[revised manuscript text omitted]

395  1), while the majority of events lead to deposition of sediments in the kettle hole surrounding flats (see Fig. 7d; Fig. 3, P1 and P4).  This statement is supported by surface runoff and sediment delivery monitoring in the study catchment (2015-2019) that has demonstrated that only very few rainfall events  caused runoff and associated sediment delivery into the kettle hole

400  (data not shown). Therefore, the study catchment shows a very limited hydrological and sedimentological connectivity between the cultivated area and kettle hole. Hence, tillage translocation in hummocky young morainic regions does also have a pronounced impact on hydrology and biogeochemistry.

**4.4 Relevance of tillage erosion and scientific attention**

Our results clearly indicate that soil erosion in the study area exceeds the tolerable soil loss rates (according to

405  Schwertmann et al., 1990 = 6 Mg ha-1 yr-1 in the study region) and is mainly attributed to tillage erosion (Fig. 6 & 7). During the socialist era (1949-1990), productivist agricultural management strategies were implemented that included land consolidation to merge large fields and the use of heavy farming machines (Forstner and Isermeyer, 2000; Wolz, 2013). For instance, annual ploughing was combined with a recommended practice of episodically using a paraplough (tillage depth ~0.6 m; Fachbereichsstandard-DDR, 1985) to break the plough pan. The average field size in the region (Quillow

[revised manuscript text omitted]

Evangeliou, N., Hamburger, T., Talerko, N., Zibtsev, S., Bondar, Y., Stohl, A., Balkanski, Y., Mousseau, T. A., and Moller, A. P.: Reconstructing the Chernobyl Nuclear Power Plant (CNPP) accident 30 years after. A unique

505      database of air concentration and deposition measurements over Europe, Environ. Pollut., 216, 408-418, 10.1016/j.envpol.2016.05.030, 2016.

Evrard, O., Chaboche, P.-A., Ramon, R., Foucher, A., and Laceby, J. P.: A global review of sediment source fingerprinting research incorporating fallout radiocesium (137Cs), Geomorphology, 362, 107103, https://doi.org/10.1016/j.geomorph.2020.107103, 2020.

510      Fachbereichsstandard-DDR: Verfahren der Pflanzenproduktion, Bodenbearbeitung, Krumenbearbeitung, TGL 28°759/03, Akademie der Landwirtschaftswissenschaften, Berlin, 1985.

Fiener, P., Dlugoß, V., and Van Oost, K.: Erosion-induced carbon redistribution, burial and mineralisation - Is the episodic nature of erosion processes important?, Catena, 133, 282-292, 10.1016/j.catena.2015.05.027, 2015.

Fiener, P., Wilken, F., Aldana-Jague, E., Deumlich, D., Gómez, J. A., Guzmán, G., Hardy, R. A., Quinton, J. N.,
515      Sommer, M., Van Oost, K., and Wexler, R.: Uncertainties in assessing tillage erosion – How appropriate are our measuring techniques?, Geomorphology, 304, 214-225, https://doi.org/10.1016/j.geomorph.2017.12.031, 2018.

Fiener, P., Wilken, F., and Auerswald, K.: Filling the gap between plot and landscape scale – eight years of soil erosion monitoring in 14 adjacent watersheds under soil conservation at Scheyern, Southern Germany, Adv.
520      Geosci., 48, 31-48, 10.5194/adgeo-48-31-2019, 2019.

Forstner, B., and Isermeyer, F.: Transformation of Agriculture in East Germany, in: Agriculture in Germany, edited by: Tangermann, S., DLG Verlag, Frankfurt a. Main, Germany, 61-90, 2000.

Gerke, H. H., Koszinski, S., Kalettka, T., and Sommer, M.: Structures and hydrologic function of soil landscapes with kettle holes using an integrated hydropedological approach, J. Hydrol., 393, 123-132,
525      10.1016/j.jhydrol.2009.12.047, 2010.

Govers, G., Vandaele, K., Desmet, P., Poesen, J., and Bunte, K.: The role of tillage in soil redistribution on hillslopes, Eur. J. Soil Sci., 45, 469-478, 10.1111/j.1365-2389.1994.tb00532.x, 1994.

Heckrath, G., Djurhuus, J., Quine, T. A., Van Oost, K., Govers, G., and Zhang, Y.: Tillage erosion and its effect on soil properties and crop yield in Denmark, J. Environ. Qual., 34, 312-324, 2005.

530      Hengl, T., and MacMillan, R. A.: Predictive soil mapping with R, OpenGeoHub foundation, Wageningen, Netherlands, 2019.

Herbrich, M., Gerke, H. H., Bens, O., and Sommer, M.: Water balance and leaching of dissolved organic and inorganic carbon of eroded Luvisols using high precision weighing lysimeters, Soil Till. Res., 165, 144-160, 10.1016/j.still.2016.08.003, 2017.

535      Hu, Y., and Kuhn, N. J.: Aggregates reduce transport distance of soil organic carbon: Are our balances correct?, Biogeosciences, 11, 6209-6219, 10.5194/bg-11-6209-2014, 2014.

Hu, Y. X., Berhe, A. A., Fogel, M. L., Heckrath, G. J., and Kuhn, N. J.: Transport-distance specific SOC distribution: Does it skew erosion induced C fluxes?, Biogeochemistry, 128, 339-351, 10.1007/s10533-016-0211-y, 2016.

540      IUSS: World reference base for soil resources 2014. Update 2015. International soil classification system for naming soils and creating legends for soil maps. World Soil Resources Reports No. 106, FAO, Rome, 2015.

Kappler, C., Kaiser, K., Tanski, P., Klos, F., Fulling, A., Mrotzek, A., Sommer, M., and Bens, O.: Stratigraphy and age of colluvial deposits indicating Late Holocene soil erosion in northeastern Germany, Catena, 170, 224-245, 10.1016/j.catena.2018.06.010, 2018.

545      Kashparov, V. A., Ahamdach, N., Zvarich, S. I., Yoschenko, V. I., Maloshtan, I. M., and Dewiere, L.: Kinetics of dissolution of Chernobyl fuel particles in soil in natural conditions, J. Environ. Radioact., 72, 335-353, https://doi.org/10.1016/j.jenvrad.2003.08.002, 2004.

Keller, T., Sandin, M., Colombi, T., Horn, R., and Or, D.: Historical increase in agricultural machinery weights enhanced soil stress levels and adversely affected soil functioning, Soil Till. Res., 194,
550      10.1016/j.still.2019.104293, 2019.

Kelley, J. M., Bond, L. A., and Beasley, T. M.: Global distribution of Pu isotopes and Np-237, Science of the Total Environment, 238, 483-500, 10.1016/s0048-9697(99)00160-6, 1999.

Ketterer, M. E., Hafer, K. M., Link, C. L., Kolwaite, D., Wilson, J., and Mietelski, J. W.: Resolving global versus local/regional Pu sources in the environment using sector ICP-MS, J. Anal. At. Spectrom., 19, 241-245, 10.1039/b302903d, 2004.

Krasa, J., Dostal, T., Jachymova, B., Bauer, M., and Devaty, J.: Soil erosion as a source of sediment and phosphorus in rivers and reservoirs – Watershed analyses using WaTEM/SEDEM, Environ. Res., 171, 470-483, https://doi.org/10.1016/j.envres.2019.01.044, 2019.

Lal, R., Ahmadi, M., and Bajracharya, R. M.: Erosional impacts on soil properties and corn yield on Alfisols in central Ohio, Land Degrad. Dev., 11, 575-585, 2000.

Li, S., Lobb, D. A., Lindstrom, M. J., and Farenhorst, A.: Tillage and water erosion on different landscapes in the northern North American Great Plains evaluated using Cs137 technique and soil erosion models, Catena, 70, 493-505, 10.1016/j.catena.2006.12.003, 2007.

Li, S., Lobb, D. A., Lindstrom, M. J., and Farenhorst, A.: Patterns of water and tillage erosion on topographically complex landscapes in the North American Great Plains, J. Soil Water Conserv., 63, 37-46, 2008.

Lust, M., and Realo, E.: Determination of dose rate from Chernobyl-derived radiocaesium in Estonian soil, J. Environ. Radioact., 112, 118-124, 10.1016/j.jenvrad.2012.05.021, 2012.

Mabit, L., Meusburger, K., Fulajtar, E., and Alewell, C.: The usefulness of 137Cs as a tracer for soil erosion assessment: A critical reply to Parsons and Foster (2011), Earth-Sci. Rev., 127, 300-307, 10.1016/j.earscirev.2013.05.008, 2013.

Mabit, L., Benmansour, M., Abril, J. M., Walling, D. E., Meusburger, K., Iurian, A. R., Bernard, C., Tarjan, S., Owens, P. N., Blake, W. H., and Alewell, C.: Fallout Pb-210 as a soil and sediment tracer in catchment sediment budget investigations: A review, Earth-Sci. Rev., 138, 335-351, 10.1016/j.earscirev.2014.06.007, 2014.

Matsunaga, T., and Nagao, S.: Environmental behavior of plutonium isotopes studied in the area affected by the Chernobyl accident, Humic Substances Research, 5/6, 19-33, 2009.

Meusburger, K., Mabit, L., Ketterer, M., Park, J. H., Sandor, T., Porto, P., and Alewell, C.: A multi-radionuclide approach to evaluate the suitability of Pu239+240 as soil erosion tracer, Science of the Total Environment, 566, 1489-1499, 10.1016/j.scitotenv.2016.06.035, 2016.

Montanarella, L., Pennock, D. J., McKenzie, N., Badraoui, M., Chude, V., Baptista, I., Mamo, T., Yemefack, M., Aulakh, M. S., Yagi, K., Hong, S. Y., Vijarnsorn, P., Zhang, G. L., Arrouays, D., Black, H., Krasilnikov, P., Sobocka, J., Alegre, J., Henriquez, C. R., Mendonca-Santos, M. D., Taboada, M., Espinosa-Victoria, D., AlShankiti, A., AlaviPanah, S. K., Elsheikh, E. A. E., Hempel, J., Arbestain, M. C., Nachtergaele, F., and Vargas, R.: World's soils are under threat, Soil, 2, 79-82, 10.5194/soil-2-79-2016, 2016.

Muramatsu, Y., Ruhm, W., Yoshida, S., Tagami, K., Uchida, S., and Wirth, E.: Concentrations of Pu-239 and Pu-240 and their isotopic ratios determined by ICP-MS in soils collected from the Chernobyl 30-km zone, Environ. Sci. Technol., 34, 2913-2917, 10.1021/es0008968, 2000.

[revised manuscript text omitted]

700    Figure 1.

[Figure]

Figure 4: Distribution histogram of 5 m x 5 m interpolated $^{239+240}$Pu measurements in 20 classes with descriptive statistics.

[Figure]

[Figure]

705

Figure 5: Linear correlation between measured and modelled $^{239+240}$Pu inventories redistributed by water ($kk_{tc}tc$:

150, P-factor: 1) and tillage erosion ($k_{til}$: 350 kg m$^{-1}$; * = p-value < 0.001). (a) Point by point correlation on 5 m x

5 m resolution (n: 1699); (b) class aggregation according to $^{239+240}$Pu derived soil redistribution. Minimum and

maximum class n is 27 and 184, respectively. While the points and classes are calculated for a reference of

710   43 Bq m$^{-2}$, the trend lines display the offset sensitivity of different reference $^{239+240}$Pu activities.

[Figure]

Figure 6: Inverse modelling of tillage and water erosion compared to $^{239+240}$Pu derived soil redistribution. Three parameter combinations (tillage transport coefficient, $k_{til}$; water transport capacity coefficient, $k_{tc}$; deviation in water erosion strength compared to reference parameterisation) are tested on their effect on the goodness-of-fit, represented by the MEF (model efficiency coefficient: perfect model fit = 1; model prediction as good as the mean = 0; model prediction worse than mean = <0).

[Figure]

Figure 7: The Figure consists of four parts: (a) Soil redistribution derived from $^{239+240}$Pu top and subsoil

720 measurements using 43 Bq m$^{-2}$ as the reference inventory; (b) geostatistically interpolated soil redistribution based

on $^{239+240}$Pu point measurements; (c) modelled tillage erosion with a tillage transport coefficient ($k_{til}$) of 350 kg m$^{-1}$;

(d) modelled water erosion according to reference parameterization ($ktc$: 150; also see Table 1).

---

## Author Response (AR2)

**Point-by-point response to comments from Kristof van Oost**

**Understanding the role of water and tillage erosion from $^{239+240}$Pu tracer measurements using inverse modelling**

Florian Wilken, Michael Ketterer, Sylvia Koszinski, Michael Sommer and Peter Fiener

We appreciate the input from Kristof van Oost and answered the comments and revised the manuscript accordingly. Please see the detailed answers (in italics) to the comments below:

**1. It appears that the vertical redistribution of PU was not modeled but its effects are still in the measurements. Can you consider in the discussion (e.g. section 4.1) the possible effect of processes like bioturbation and colloid migration on the vertical distribution of PU?**

*Thanks, for this comment! The model calculates and keeps track of the vertical redistribution of $^{239+240}$Pu along the soil profile. However, this depth explicit information was not used to calculate soil redistribution. The soil redistribution calculation of both measurement and modelling data is based on the change of the inventory. Therefore, the key advantage of modelling the vertical $^{239+240}$Pu profile development is to include depletion processes at eroding sites and associated deposition of depleted sediments. As the calculations are based on inventories, a pronounced effect of bioturbation and colloid migration would require transport below the sampling depth. This might partly take place, however, Fig. 3 (measurements of high resolution depth distribution) shows a rapid decrease of the $^{239+240}$Pu activity below the plough layer at eroding sites and below 45 cm depth at most depositional sites. Therefore, we can assume that the effect of bioturbation and colloid migration has a very limited effect on the soil redistribution results and vertical profile development. We include a statement on bioturbation and colloid migration in section 4.1 as follows:*

*"The application of soil tracers like fallout radionuclides are based on the assumption that spatial and vertical patterns are exclusively caused by soil redistribution processes due to water and tillage erosion. Bioturbation or colloid migration can be drivers of vertical displacement, which can cause an incomplete representation of the inventory if the downward migration exceeds the soil sampling depth. Within this study, a rapid reduction of the $^{239+240}$Pu activity was found below the plough and colluvial layer of the high resolution depth profiles (see Fig. 3). Hence, there is no indication for a pronounced downward migration of $^{239+240}$Pu below the sampling depth."*

**2. Would the model, with a vertical discretization of 10 cm, not show significant numerical dispersion?**

*Thank you for pointing at this issue. The model is to some extent subject to numerical dispersion that corresponds to the vertical resolution of the profile. However, as pointed out in our answer to point 1, we calculate soil redistribution based on changes in inventories, which limits the effect of this problem. To get a grip on the sensitivity of the model to the vertical*
*resolution, we tested a reduction of the soil layer thickness from 10 cm to 5 cm that doubled the number of soil layers. The comparison between the two vertical resolutions show a maximum difference of -1.8 Bq m$^{-2}$ and a 25% and 75% quartile of -0.12 and 0.37 Bq m$^{-2}$, respectively.*

[Figure]

Figure 1: Histogram of model difference between 10 (10 cm resolution) and 20 (5 cm resolution) soil layer vertical resolution
after a simulation period of 55 yrs. The model runs were carried out using best-fit model parameterisations (please see manuscript) and a reference inventory of 43 Bq m$^{-2}$.

*Based on this example, we conclude that the effect of the vertical resolution is minor and does not have a substantial impact on the results. This would be different if we would consider the migration of the $^{239+240}$Pu rich topsoil versus the $^{239+240}$Pu free*
*subsoil. We agree that the issue of resolution was not discussed in the manuscript sufficiently and we therefore include a discussion in the text of section 4.2 as follows:*

*"Numerical dispersion, can affect the model results, which is related to the vertical and spatial representation of the model. The vertical soil profile representation in 10 cm increments is rather low for a fallout radionuclide tracer with a rapid decrease of 239+240Pu between the plough layer/colluvium and the subsoil layer (see Fig. 3). However, the calculation of soil*
*redistribution is based on relative changes of the 239+240Pu inventory in relation to reference inventories. Therefore, the*

*sensitivity of the soil redistribution calculations on the 239+240Pu soil profile development is minor as the 239+240Pu inventory is independent from the depth distribution. The relatively high spatial resolution of 5 m x 5 m was selected to adequately represent the high landscape variability in the study area and has reliably simulated spatial patterns in previous studies (e.g. Van Oost et al., 2003; Dlugoß et al., 2012). Overall, the model performs well and explains 95% of the variance*

*and achieves a MEF better than 0.8 in predicting 239+240Pu patterns by simulating the combined effect of three model parameters (tillage translocation coefficient, water transport capacity coefficient, water erosion strength)."*

**3. Does picking the 3 parameter sets for the inverse modelling (ktil, ktc and water erosion strength) imply that the model is assumed to be perfect, and that all uncertainty (model misfit, via MEF) is attributed to parameter settings?**

**Do you consider this a strong assumption?**

*The model is subject to a range of uncertainties e.g. no topography update over the past 53 yrs., short-term topographical variation (e.g. tram lines) at the time of the LIDAR measurement have a distinct effect on the simulation (for your information: we addressed this issue by applying a low pass filter), the use of a spatially homogeneous $k_{til}$, which might be topography related. We do not assume all of the uncertainty goes back to the parameter settings. Nevertheless, the model is able the explain*

*up to 95% of variance (aggregated results; 45% on raster level), which is from our perspective excitingly high. To point out potential uncertainties that are related to the model, we extended the uncertainty section (4.2) as follows:*

*"Within this study, an inverse modelling approach was carried out to understand the contribution of soil redistribution by water and tillage erosion. The model is subject to specific uncertainties that need to be mentioned. The diffusion type equation for the tillage erosion simulation (Govers et al., 1994) follows the assumption of a spatially homogenous tillage translocation*

*coefficient for the entire study catchment. However, as the tillage translocation coefficient is a function of tillage speed and depth, tillage translocation distances are likely to show spatial differences that may follow the topography of the study area. Furthermore, the model uses a static digital elevation model that does not account for topographical change during the simulation period, which causes static soil redistribution patterns that ignore feedback processes. This is an issue for the used digital elevation model that represents short-term topographical structures like agricultural tramlines at the acquisition time.*

*To reduce this effect, a low pass filter was applied on the digital elevation model to somewhat even out short-term structures."*

In this study, we will determine the soil redistribution patterns in a small 4.2 ha catchment based on high-resolution $^{239+240}$Pu measurements and analyse the contribution of water and tillage erosion processes based on an inverse modelling approach using a combined water and tillage erosion model. The general  objective is to unravel the importance of water and tillage erosion driving the current variability of soil properties in an intensively used arable landscape of north-eastern Germany.

**2. Methods**

**2.1 Study area**

The study area (53°21'2 N, 13°39'5 E) is situated in the hummocky ground morainic landscape of the Weichselian glacial belt ('young morainic area') of north-eastern Germany (Fig. 1). Characteristic for these landscapes are widespread closed depressions, so-called kettle holes, which result from a delayed melting of dead ice blocks. They are nowadays filled with mineral soil, (degraded) peat or water. The study area is part of a kettle hole catchment (4.2 ha; Fig. 1)  showing a high morphological variability covering convex hilltops, steep slopes and flat areas . The recent crop rotation is rape (*Brassica napus* L.)-winter wheat (*Triticum aestivum* L.)-winter barley (*Hordeum vulgare* L.)-winter barley, cultivated without cover crops, which is a typical conventional crop rotation that is adapted for the highly fertile soils of the Uckermark region. The mean arable land of a farm in the region is 352 ha, which is much larger compared to the mean of the State of Brandenburg (250 ha) and Germany (60 ha; Troegel and Schulz, 2018). These larger field sizes are explained by land consolidation programmes implemented in the 1960s during the socialist period. However, also before, agriculture in the region was already characterised by large scale farming structures  and corresponding high agricultural mechanisation. In 1939, large farms that manage more than 100 ha of arable land cultivated 7% of the total arable area of Germany. In contrast, within the present-day federal states of Mecklenburg-Pomerania, Brandenburg (study area location) and Saxony-Anhalt, large farms cultivated 30% of corresponding arable land (Wolz, 2013). The catchment is part of a single large field (54 ha), which is a size that can be frequently found in the region. The soils are developed from glacial till and vary with respect to their location in the landscape. Convex hilltops and steep slopes are dominated by extremely eroded A-C profiles (Calcaric Regosols, soil classification according to: IUSS, 2015), while Luvisols showing different degrees of erosion that are typically situated at the up and mid slopes, the footslopes and depressions  are dominated by Gleyic-Colluvic Regosols (Fig. 1b; Sommer et al., 2008; Gerke et al., 2010). These soils regularly reveal fossil surface horizons below 1 m depth (fAh, fH). 
[revised manuscript text omitted]

To understand the drivers of current $^{239+240}$Pu and associated soil degradation patterns, an inverse modelling was carried out that was  quantitatively analysed by goodness-of-fit parameters. The spatial correlation between the $^{239+240}$Pu derived patterns and the modelled best knowledge soil redistribution, including both water and tillage erosion, is only moderate (R$^2$ =  0.45, Rho = 0.73) on a raster by raster comparison (n =  1699, 5 m x 5 m grid points; Fig. 5a).  To reduce  small-scale variability and understand the average goodness-of-fit, the inverse modelling results were classified according to the measured $^{239+240}$Pu activity. The classified results average out the spatio-temporal dynamics and show a very high correlation (R$^2$ =  0.95, Rho = 0.99; Fig. 5b), which  illustrates the great agreement of the spatial soil redistribution patterns between the $^{239+240}$Pu measurements and the model results.

While the analysis of the spatial correlation is a relative comparison, the absolute deviation is considered according to the MEF (model efficiency coefficient; Nash and Sutcliffe, 1970; 1 = perfect prediction, 0 = as good as mean of all measurements, < 0 = worse than mean). The quality of model predictions shows hardly any sensitivity to water erosion related parameterisations ($k_{tc}$ and erosion strength; Figure 6c). In contrast, the tillage erosion strength, represented by $k_{til}$

parameter iterations, showed a substantial impact on the MEF (Fig. 6a & b). A MEF better than 0.8 and RMSE below 6.5 Bq m$^{-2}$ were found for a $k_{til}$ range from 225 to 475 kg m$^{-1}$, while the best model fit was found for a $k_{til}$ of 350 kg m$^{-1}$ achieving a MEF of 0.87 and a corresponding RMSE of 5.2 Bq m$^{-2}$. The best model fit was found without the contribution of water erosion. The highest impact on the best-fit model run was a 0.31 MEF-reduction by an extreme water erosion parameterisation ($k_{tc}$ = 500, water erosion strength = 200%).

Soil redistribution determined by the proportional conversion approach using $^{239+240}$Pu measurements, indicates substantial morphodynamics geomorphological dynamics in the study catchment over the past decades. Soil erosion at hilltop locations is shown to be reach up to 14.9 cm (43 Bq m$^{-2}$ reference; 40 Bq m$^{-2}$ reference = 14.1 cm; 46 Bq m$^{-2}$ reference = 15.6 cm), while deposition partly can builds a colluvium layer up with a maximum thickness of 21.5 cm (43 Bq m$^{-2}$ reference; 40 Bq m$^{-2}$ reference = 24.9 cm; 46 Bq m$^{-2}$ reference = 18.6 cm) colluvium over the past 53 yr. The inverse modelling stresses that substantial soil erosion, which takes place over large areas, is almost exclusively attributed to tillage translocation (modelled max. water erosion = 3.8 cm (53 yr)$^{-1}$ vs. max. tillage erosion = 13.5 cm (53 yr)$^{-1}$; Fig. 7c, d). In turn, both processes contribute to deposition in the kettle hole surrounding flats (max. water deposition = 27.1 cm (53 yr)$^{-1}$ vs. max. tillage deposition = 22.4 cm (53 yr)$^{-1}$; Fig. 7c, d).

**4 Discussions**

**4.1 $^{239+240}$Pu methodological benefits and limitations**

The use of fallout radionuclides to determine soil redistribution patterns and rates over the past decades has been used in many studies in various study areas around the world (see reviews: Mabit et al., 2014; Alewell et al., 2017; Evrard et al., 2020) and contributed substantially to understand soil degradation processes. However, the most frequently used fallout radionuclide $^{137}$Cs faces upcoming limitations (Chernobyl fallout that adds on the global fallout over large parts of Europe and ongoing decay below detection limit of standard measuring devices; also see section 1) in the use as a soil redistribution tracer (Evrard et al., 2020). The fallout radionuclide $^{239+240}$Pu has demonstrated its suitability to determine the recent soil redistribution history (since the 1960s; see review Alewell et al., 2017) and is a potential alternative for fills the gap of upcoming limitations in the use of $^{137}$Cs as a soil redistribution tracer (Mabit et al., 2013; Alewell et al., 2017). In Europe, where large parts were re-contaminated by $^{137}$Cs fallout of the Chernobyl accident (Evangeliou et al., 2016), additional information on the spatial change on the inventory is needed to derive accurate soil redistribution rates. Particularly in the area of the former GDR, almost no information that can be used for a correction on the $^{137}$Cs Chernobyl re-contamination are available (Evangeliou et al., 2016). The $^{239+240}$Pu fallout caused by the Chernobyl disaster was very local (radius <100 km) and has a distinct fingerprint based on the $^{239}$Pu/$^{240}$Pu ratio. While the $^{240}$Pu/$^{239}$Pu ratio of global fallout in the Northern Hemisphere is 0.180±0.014 (Kelley et al., 1999), the $^{240}$Pu/$^{239}$Pu ratio soils that received high Chernobyl fallout is about twice as high (0.408±0.003, determined for soils within the 30 km exclusion zone of the Chernobyl reactor; Muramatsu et al., 2000; Boulyga and Becker, 2002). The 95% interval of confidence and average of the $^{240}$Pu/$^{239}$Pu ratio found in the soil samples of this study were 0.281 and 0.199, respectively. Hence, a relevant $^{239+240}$Pu re-contamination by Chernobyl fallout can be ruled out for the study area. Another limitation for the use of $^{137}$Cs as a soil redistribution tracer is the ongoing decay due to short half-life times that has already caused a substantial reduction of the inventory. Due to lower activities, measuring devices of much higher complexity are needed in the future (Evrard et al., 2020). Decay is not an issue for $^{239}$Pu and $^{240}$Pu as both nuclides have long half-life times that allow for a quasi-unlimited use, however, it needs to be mentioned that sample preparation for $^{239+240}$Pu ICP-MS

measurements is much more laborious compared to the standard procedure of physical $^{137}$Cs measurements. However, some methodological limitations persist that need to be taken into account.

Enrichment processes, due to selective transport of soil constituents that fallout radionuclides are preferentially associated with, are a critical issue for the use of most (e.g., $^{239+240}$Pu, $^{137}$Cs, $^{210}$Pb) radionuclide tracers (Parsons and Foster, 2011; Mabit et al., 2014; Alewell et al., 2017). While $^{137}$Cs is mainly associated with clay particles, $^{239+240}$Pu binds to soil organic matter and oxides (Alewell et al., 2017) that are less affected by selective water transport and corresponding $^{239+240}$Pu enrichment (Meusburger et al., 2016; Xu et al., 2017). However, it needs to be mentioned that radionuclide associated particles are typically not transported as primary particles but in soil aggregate complexes (Hu and Kuhn, 2014; Hu et al., 2016), which has a pronounced effect on enrichment processes (Wilken et al., 2017b). Nevertheless, the $^{239+240}$Pu activity at depositional sites, that are redistributed by water (transport by tillage is typically assumed to be non-grain size selective; Fiener et al., 2018), can be higher in relation to the activity of the source material. A soil profile that shows a distinct indicator of enrichment processes in this study is sampling profile P1 (25-45 cm; Fig. 3) that is situated in the kettle hole surrounding flat. Hence, enrichment in fine particles of relatively high $^{239+240}$Pu activity is to some extent also an issue within this study that causes an overestimation of deposition. A particle size correction factor was not applied as topsoil enrichment (topsoil Bq m$^{-2}$ > ref. 43 Bq m$^{-2}$) was exclusively found at very few sampling locations (<6%) in the kettle hole surrounding flats. Furthermore, the mean topsoil ratio of enriched sediments is moderate (1.2) and supports the general assumption that $^{239+240}$Pu is less affected by selective transport compared to $^{137}$Cs (Alewell et al., 2017) and that transport by tillage is non-grain size specific. The counteracting process of enrichment is the deposition of $^{239+240}$Pu depleted sediments that are transported from highly eroded locations. Such highly depleted locations can be found at the hilltops of the study area (Fig. 7b). Hence, the hilltops are the main source of highly depleted sediments that are deposited in kettle hole surrounding flats. However, the minimum horizontal distance from the hilltops to the kettle hole surrounding flat is roughly about 70 m and the approximate tillage translocation distance 0.5 to 1 m per pass (Fiener et al., 2018). Hence, deposition of depleted $^{239+240}$Pu material has to be mainly attributed to surface runoff that can bridge flow across longer transport distances. SPEROS-Pu takes depletion of deposited sediments into account but does not address enrichment processes. Furthermore, a maximum soil sampling depth down to 50 cm was carried out within this study that technically allows to derive a maximum depositional depth of 25 cm using the proportional conversion approach of Walling et al. (2011), which is at four sampling locations exceeded was exceeded at four sampling locations. Nevertheless, also with a deeper soil sampling, it would be arguable if these potentially enriched or depleted sampling locations should be excluded from the statistical analysis, like it was done for extreme depositional locations (4 sampling locations) within this study.

The application of soil tracers like fallout radionuclides are based on the assumption that spatial and vertical patterns are
exclusively caused by soil redistribution processes due to water and tillage erosion. Bioturbation or colloid migration can be
drivers of vertical displacement, which can cause an incomplete representation of the inventory if the downward migration
exceeds the soil sampling depth. Within this study, a rapid reduction of the $^{239+240}$Pu activity was found below the plough and
colluvial layer of the high resolution depth profiles (see Fig. 3). Hence, there is no indication for a pronounced downward
migration of $^{239+240}$Pu below the sampling depth.

**4.2 Using $^{239+240}$Pu and inverse modelling to understand the recent soil erosion history**

Within this study, an inverse modelling approach was carried out to understand the contribution of soil redistribution by water
and tillage erosion. The model is subject to specific uncertainties that need to be mentioned. The diffusion type equation for
the tillage erosion simulation (Govers et al., 1994) follows the assumption of a spatially homogenous tillage translocation
coefficient for the entire study catchment. However, as the tillage translocation coefficient is a function of tillage speed and
depth, tillage translocation distances are likely to show spatial differences that may follow the topography of the study area.
Furthermore, the model uses a static digital elevation model that does not account for topographical change during the
simulation period, which causes static soil redistribution patterns that ignore feedback processes. This is an issue for the used
digital elevation model that represents short-term topographical structures like agricultural tramlines at the acquisition time.
To reduce this effect, a low pass filter was applied on the digital elevation model to somewhat even out short-term structures.
Numerical dispersion, can affect the model results, which is related to the vertical and spatial representation of the model. The
vertical soil profile representation in 10 cm increments is rather low for a fallout radionuclide tracer with a rapid decrease of
$^{239+240}$Pu between the plough layer/colluvium and the subsoil layer (see Fig. 3). However, the calculation of soil redistribution
is based on relative changes of the $^{239+240}$Pu inventory in relation to reference inventories. Therefore, the sensitivity of the soil
redistribution calculations on the $^{239+240}$Pu soil profile development is minor as the $^{239+240}$Pu inventory is independent from the
depth distribution. The relatively high spatial resolution of 5 m x 5 m was selected to adequately represent the high landscape
variability in the study area and has reliably simulated spatial patterns in previous studies (e.g. Van Oost et al., 2003; Dlugoß
et al., 2012). Overall, the model performs well and explains 95% of the variance and achieves a MEF better than 0.8 in
predicting $^{239+240}$Pu patterns by simulating the combined effect of three model parameters (tillage translocation coefficient,
water transport capacity coefficient, water erosion strength).
Within the intensively used managed study catchment, substantial $^{239+240}$Pu derived soil redistribution was found with soil loss
up to 45 Mg ha$^{-1}$ yr$^{-1}$ (ref. 43 Bq m$^{-2}$; ref. 40 and 46 Bq m$^{-2}$ = 43 and 47 Mg ha$^{-1}$ yr$^{-1}$) and sediment deposition up to 65 Mg
ha$^{-1}$ yr$^{-1}$ (ref. 43 Bq m$^{-2}$; ref. 40 and 46 Bq m$^{-2}$ = 75 and 56 Mg ha$^{-1}$ yr$^{-1}$). Very high deposition can only be found in the
spatially narrow area of the kettle hole surrounding flat where both water and tillage erosion processes water and tillage erosion
lead to deposition (Fig. 1). The kettle hole surrounding flat is a spatially narrow area, but the only region zone where water
erosion substantially contributes to pronounced morphodynamics geomorphological dynamics (Fig. 7d). As a result of the
small spatial contribution extent where this process takes place, the inverse modelling shows hardly any sensitivity on goodness-of-fit changes in reaction to the variation in model parameterisations (Fig. 6c). Nevertheless, sediment deposition and delivery by surface runoff is a important process in the study area. Evidence for runoff-based sediment delivery is a colluvial layer covering the peat in the kettle hole with an average depth of 40 cm. This sediment delivery into the kettle hole cannot be explained by the inverse modelling of water erosion applying a reasonable parameter range . Therefore, we assume the reference parameterisation for the region given by the state of Brandenburg  as the most appropriate ($k_{tc}$ : 150 m and RUSLE parameters according to Tab. 1). According to the model run using the reference parameterisation for water erosion, a colluvial layer of 1.7 cm (53 yr)$^{-1}$ would have been developed on top of the peat that has been exported from the arable part of the catchment (see Fig. 1) due to water transport over the past decades. This indicates a long water erosion history before the 1960s. This is not surprising as bare soil conditions and erosive rainfall events have taken place since the onset of arable use approximately 1 k yr before present (Van der Meij et al., 2017; Kappler et al., 2018). In contrast, tillage erosion is typically assumed to be a process that is linked to recent developments of increasing mechanical forces that have been applied to soils over the past century (Sommer et al., 2008; Calitri et al., 2019). Within this study, a maximum topographical change by hilltop erosion up to 17 cm (53 yr)$^{-1}$ was determined. In a review on tillage erosion by Van Oost and Govers (2006), tillage translocation coefficients of 44 experiments were reported for different tillage practices. This resulted in a mean $k_{til}$ of 234 kg m$^{-1}$ ( 5% percentile : 30 kg m$^{-1}$; 95% percentile : 640 kg m$^{-1}$) for mouldboard and chisel plough. Within this study, a tillage translocation coefficient of 350 kg m$^{-1}$ per year was determined. The tillage translocation coefficients, determined by the inverse modelling approach, are rather high compared to other studies considering that fallout radioisotopic tracer approaches  cover a phase of high mechanical development from low to high power farming machines (Sommer et al., 2008; Keller et al., 2019). Although recent tillage translocation rates are rather high they cannot explain the soil depth patterns that are visible by augerings. In the study region, it can be observed that tillage erosion mainly affected hilltops. Calcaric glacial till is approaching the surface by soil profile truncation and is partially mixed into the plough layer. Within the study catchment, this is the case for 20 sampling locations (CaCO$_3$ > 0.5%) that are also indicated as  the most eroded sites by the $^{239+240}$Pu measurements and the inverse modelling. Non-eroded reference profiles (n  ÷210) in the region show the parent material (calcaric glacial till) is found at 102 cm depth on average (van der Meij et al., 2019). Hence, less than 17% of soil depth reduction can be attributed to most recent process dynamics. This suggests that traditional hand or cattle based tillage systems, which are used since the beginning of arable agriculture in the region (1k yr BP; Kappler et al., 2018), must have caused extensive soil redistribution over long periods. This suggests that tillage erosion might be the dominant process even without mechanized soil tillage, which is the common practice in most developing countries that also partly cultivate very steep slopes. Therefore, the general assumption of tillage erosion being only an issue for highly mechanised agricultural systems (Van Oost et al., 2006) might need to be reconsidered across a range of contrasted agricultural environments.

**4.3 Interplay of sediment redistribution by water and tillage**

The inverse modelling has shown that soil redistribution by water has only a minor impact on erosion processes in the study area. However, sediment deposition by water has a complex interplay with tillage translocation (kettle hole surrounding flat; Fig. 7). Very high deposition by tillage translocation towards the field-kettle hole edge (typically >1 m known from soil augering; Kappler et al., 2018) builds up local hydrological depressions (Fig. 1 & 7). Only infrequent extreme events exceed the critical runoff quantity to connect the arable hillslopes with the inner peat area of  the kettle hole (Fig.

1), while the majority of events  lead to deposition of sediments in the kettle hole surrounding flats (see Fig. 7d; Fig. 3, P1 and P4).  This statement is supported by surface runoff and sediment delivery monitoring in the study catchment (2015-2019)  that has demonstrated that only very few rainfall events  caused runoff and associated sediment delivery into the kettle hole (data not shown). Therefore, the study catchment shows a very limited hydrological and sedimentological connectivity between the cultivated area and kettle hole. Hence, tillage translocation in hummocky young morainic regions does also have a pronounced impact on hydrology and biogeochemistry.

**4.4 Relevance of tillage erosion and scientific attention**

Our results clearly indicate that soil erosion in the study area exceeds the tolerable soil loss rates (according to

Schwertmann et al., 1990 = 6 Mg ha-1 yr-1 in the study region) and is mainly attributed to tillage erosion (Fig. 6 & 7). During the socialist era (1949-1990),  productivist agricultural management strategies were implemented that included land  consolidation to merge large fields and the use of heavy farming machines (Forstner and Isermeyer, 2000; Wolz, 2013). For instance, annual ploughing was combined with a recommended practice of episodically using a paraplough (tillage depth ~0.6 m; Fachbereichsstandard-DDR, 1985) to break the plough pan. The average field size in the region (Quillow catchment : 22 ha) is rather large, this  has  favoured big farming structures that utilises  powerful machinery. However, tillage erosion does not receive reasonable scientific attention (Fiener et al., 2018), even if  its effects on yields (Oettl et al., 2020), nutrient and carbon cycling (Wilken et al., 2017a; Zhao et al., 2018; Nie et al., 2019) and soil hydrology (Herbrich et al., 2017) are widely known. Globally, tillage erosion has been recognized as an environmental threat in the  hummocky young morainic regions that have shallow soils that are subject to dropping yields at hilltop locations  (Canada: Pennock, 2003; Tiessen et al., 2007a; Tiessen et al., 2007b, North America: Li et al., 2007, 2008, Russia: Olson et al., 2002; Belyaev et al., 2005 and Northern Europe: Quine et al., 1994; Heckrath et al., 2005; Wysocka-Czubaszek and Czubaszek, 2014) . Most arable regions are subject to pronounced tillage erosion (e.g. illustrated in the landscape by tillage banks along downslope field borders; Chartin et al., 2013), but may not show a pronounced impact on yields (Lal et al., 2000). Loess derived soils with a homogeneous grain size distribution for several meters of depth do not show major differences in soil structure (Blume et al., 2016), while nutrient losses are compensated by fertilizer applications. Another reason for not being a prominent soil degradation mechanism might be that the impacts of tillage erosion  are  not as visible as those caused by  water erosion, which leads to rapid topographical dynamics (rills and gullys) and off-site damages (muddy floods, siltation). However, tillage erosion is a highly important soil redistribution process, taking place on the majority of sloped arable fields that urgently needs scientific consideration  and implementation in soil conservation management by policy makers.

**5 Conclusions**

In this study, $^{239+240}$Pu was used as a tracer to  reconstruct soil redistribution in a hummocky young morainic study catchment under intense arable use. To understand the role of water and tillage erosion on soil degradation patterns, an inverse modelling approach was carried out in the study catchment. The results clearly show that recent soil degradation in the study area is dominated by tillage translocation. Furthermore, tillage erosion has a substantial impact on surface  runoff. Tillage forms hydrological depressions at the downslope border between the cultivated field and the kettle hole that limits the  surface runoffhydrological and sedimentological connectivity into the kettle hole and  causes deposition of sediments that are transported by water. Soil redistribution by water has no major contribution to soil loss  on the catchment hillslopes, but causes pronounced deposition in the spatially narrow area of the kettle hole surrounding flat. Within this study, soil erosion up to 17 cm (53 yr)$^{-1}$ and deposition exceeding 25 cm (53 yr)$^{-1}$ of recent  geomorphological dynamics (since 1960s) were found. However, even these relatively high erosion rates cannot explain the current soil degradation patterns determined from soil prospection and chemical analysis that show both profile soil truncation and colluviation larger than one meter. This indicates that tillage erosion might  not be a process that exclusively takes place in highly mechanised agro-ecosystems but is potentially causing pronounced soil degradation in smallholder farming structures. Our results clearly underline that tillage erosion is a critically underrepresented soil degradation process that can be the main soil redistribution driver on catchment scale.

**Data availability**

The data will be made available on request.

**Author contribution**

[revised manuscript text omitted]

Fachbereichsstandard-DDR: Verfahren der Pflanzenproduktion, Bodenbearbeitung, Krumenbearbeitung, TGL 28°759/03, Akademie der Landwirtschaftswissenschaften, Berlin, 1985.

Fiener, P., Dlugoß, V., and Van Oost, K.: Erosion-induced carbon redistribution, burial and mineralisation - Is the episodic nature of erosion processes important?, Catena, 133, 282-292, 10.1016/j.catena.2015.05.027, 2015.

Fiener, P., Wilken, F., Aldana-Jague, E., Deumlich, D., Gómez, J. A., Guzmán, G., Hardy, R. A., Quinton, J. N., Sommer, M., Van Oost, K., and Wexler, R.: Uncertainties in assessing tillage erosion – How appropriate are our measuring techniques?, Geomorphology, 304, 214-225, https://doi.org/10.1016/j.geomorph.2017.12.031, 2018.

Fiener, P., Wilken, F., and Auerswald, K.: Filling the gap between plot and landscape scale – eight years of soil erosion monitoring in 14 adjacent watersheds under soil conservation at Scheyern, Southern Germany, Adv. Geosci., 48, 31-48, 10.5194/adgeo-48-31-2019, 2019.

Forstner, B., and Isermeyer, F.: Transformation of Agriculture in East Germany, in: Agriculture in Germany, edited by: Tangermann, S., DLG Verlag, Frankfurt a. Main, Germany, 61-90, 2000.

Gerke, H. H., Koszinski, S., Kalettka, T., and Sommer, M.: Structures and hydrologic function of soil landscapes with kettle holes using an integrated hydropedological approach, J. Hydrol., 393, 123-132, 10.1016/j.jhydrol.2009.12.047, 2010.

Govers, G., Vandaele, K., Desmet, P., Poesen, J., and Bunte, K.: The role of tillage in soil redistribution on hillslopes, Eur. J. Soil Sci., 45, 469-478, 10.1111/j.1365-2389.1994.tb00532.x, 1994.

Heckrath, G., Djurhuus, J., Quine, T. A., Van Oost, K., Govers, G., and Zhang, Y.: Tillage erosion and its effect on soil properties and crop yield in Denmark, J. Environ. Qual., 34, 312-324, 2005.

Hengl, T., and MacMillan, R. A.: Predictive soil mapping with R, OpenGeoHub foundation, Wageningen, Netherlands, 2019.

Herbrich, M., Gerke, H. H., Bens, O., and Sommer, M.: Water balance and leaching of dissolved organic and 560 inorganic carbon of eroded Luvisols using high precision weighing lysimeters, Soil Till. Res., 165, 144-160, 10.1016/j.still.2016.08.003, 2017.

Hu, Y., and Kuhn, N. J.: Aggregates reduce transport distance of soil organic carbon: Are our balances correct?, Biogeosciences, 11, 6209-6219, 10.5194/bg-11-6209-2014, 2014.

Hu, Y. X., Berhe, A. A., Fogel, M. L., Heckrath, G. J., and Kuhn, N. J.: Transport-distance specific SOC 565 distribution: Does it skew erosion induced C fluxes?, Biogeochemistry, 128, 339-351, 10.1007/s10533-016-0211-y, 2016.

IUSS: World reference base for soil resources 2014. Update 2015. International soil classification system for naming soils and creating legends for soil maps. World Soil Resources Reports No. 106, FAO, Rome, 2015.

Kappler, C., Kaiser, K., Tanski, P., Klos, F., Fulling, A., Mrotzek, A., Sommer, M., and Bens, O.: Stratigraphy and 570 age of colluvial deposits indicating Late Holocene soil erosion in northeastern Germany, Catena, 170, 224-245, 10.1016/j.catena.2018.06.010, 2018.

Kashparov, V. A., Ahamdach, N., Zvarich, S. I., Yoschenko, V. I., Maloshtan, I. M., and Dewiere, L.: Kinetics of dissolution of Chernobyl fuel particles in soil in natural conditions, J. Environ. Radioact., 72, 335-353, https://doi.org/10.1016/j.jenvrad.2003.08.002, 2004.

Keller, T., Sandin, M., Colombi, T., Horn, R., and Or, D.: Historical increase in agricultural machinery weights enhanced soil stress levels and adversely affected soil functioning, Soil Till. Res., 194, 10.1016/j.still.2019.104293, 2019.

Kelley, J. M., Bond, L. A., and Beasley, T. M.: Global distribution of Pu isotopes and Np-237, Science of the Total Environment, 238, 483-500, 10.1016/s0048-9697(99)00160-6, 1999.

Ketterer, M. E., Hafer, K. M., Link, C. L., Kolwaite, D., Wilson, J., and Mietelski, J. W.: Resolving global versus local/regional Pu sources in the environment using sector ICP-MS, J. Anal. At. Spectrom., 19, 241-245, 10.1039/b302903d, 2004.

Krasa, J., Dostal, T., Jachymova, B., Bauer, M., and Devaty, J.: Soil erosion as a source of sediment and phosphorus in rivers and reservoirs – Watershed analyses using WaTEM/SEDEM, Environ. Res., 171, 470-483, 585 https://doi.org/10.1016/j.envres.2019.01.044, 2019.

Lal, R., Ahmadi, M., and Bajracharya, R. M.: Erosional impacts on soil properties and corn yield on Alfisols in central Ohio, Land Degrad. Dev., 11, 575-585, 2000.

Li, S., Lobb, D. A., Lindstrom, M. J., and Farenhorst, A.: Tillage and water erosion on different landscapes in the northern North American Great Plains evaluated using Cs137 technique and soil erosion models, Catena, 70, 493-590 505, 10.1016/j.catena.2006.12.003, 2007.

Li, S., Lobb, D. A., Lindstrom, M. J., and Farenhorst, A.: Patterns of water and tillage erosion on topographically complex landscapes in the North American Great Plains, J. Soil Water Conserv., 63, 37-46, 2008.

Lust, M., and Realo, E.: Determination of dose rate from Chernobyl-derived radiocaesium in Estonian soil, J. Environ. Radioact., 112, 118-124, 10.1016/j.jenvrad.2012.05.021, 2012.

Mabit, L., Meusburger, K., Fulajtar, E., and Alewell, C.: The usefulness of 137Cs as a tracer for soil erosion assessment: A critical reply to Parsons and Foster (2011), Earth-Sci. Rev., 127, 300-307, 10.1016/j.earscirev.2013.05.008, 2013.

Mabit, L., Benmansour, M., Abril, J. M., Walling, D. E., Meusburger, K., Iurian, A. R., Bernard, C., Tarjan, S., Owens, P. N., Blake, W. H., and Alewell, C.: Fallout Pb-210 as a soil and sediment tracer in catchment sediment 600 budget investigations: A review, Earth-Sci. Rev., 138, 335-351, 10.1016/j.earscirev.2014.06.007, 2014.

Matsunaga, T., and Nagao, S.: Environmental behavior of plutonium isotopes studied in the area affected by the Chernobyl accident, Humic Substances Research, 5/6, 19-33, 2009.

Meusburger, K., Mabit, L., Ketterer, M., Park, J. H., Sandor, T., Porto, P., and Alewell, C.: A multi-radionuclide approach to evaluate the suitability of Pu239+240 as soil erosion tracer, Science of the Total Environment, 566, 605 1489-1499, 10.1016/j.scitotenv.2016.06.035, 2016.

Montanarella, L., Pennock, D. J., McKenzie, N., Badraoui, M., Chude, V., Baptista, I., Mamo, T., Yemefack, M., Aulakh, M. S., Yagi, K., Hong, S. Y., Vijarnsorn, P., Zhang, G. L., Arrouays, D., Black, H., Krasilnikov, P., Sobocka, J., Alegre, J., Henriquez, C. R., Mendonca-Santos, M. D., Taboada, M., Espinosa-Victoria, D., AlShankiti, A., AlaviPanah, S. K., Elsheikh, E. A. E., Hempel, J., Arbestain, M. C., Nachtergaele, F., and Vargas, 610 R.: World's soils are under threat, Soil, 2, 79-82, 10.5194/soil-2-79-2016, 2016.

Muramatsu, Y., Ruhm, W., Yoshida, S., Tagami, K., Uchida, S., and Wirth, E.: Concentrations of Pu-239 and Pu-240 and their isotopic ratios determined by ICP-MS in soils collected from the Chernobyl 30-km zone, Environ. Sci. Technol., 34, 2913-2917, 10.1021/es0008968, 2000.

[revised manuscript text omitted]

Figure 5: Linear correlation between measured and modelled $^{239+240}$Pu inventories redistributed by water ($\cancel{kk_{tc}\cancel{tc}}$:
150, P-factor: 1) and tillage erosion ($k_{til}$: 350 kg m$^{-1}$; * = p-value < 0.001). (a) Point by point correlation on 5 m x
m resolution (n: 1699); (b) class aggregation according to $^{239+240}$Pu derived soil redistribution. Minimum and
maximum class n is 27 and 184, respectively. While the points and classes are calculated for a reference of
Bq m$^{-2}$, the trend lines display the offset sensitivity of different reference $^{239+240}$Pu activities.

[Figure]

Figure 6: Inverse modelling of tillage and water erosion compared to $^{239+240}$Pu derived soil redistribution. Three parameter combinations (tillage transport coefficient, $k_{til}$; water transport capacity coefficient, $k_{tc}$; deviation in water erosion strength compared to reference parameterisation) are tested on their effect on the goodness-–of-–fit, represented by the MEF (model efficiency coefficient: perfect model fit = 1; model prediction as good as the mean = 0; model prediction worse than mean = <0).

[Figure]

Figure 7: The Figure consists of four parts: (a) Soil redistribution derived from $^{239+240}$Pu top and subsoil measurements using 43 Bq m$^{-2}$ as the reference inventory; (b) geostatistically interpolated soil redistribution based on $^{239+240}$Pu point measurements; (c) modelled tillage erosion with a tillage transport coefficient ($k_{til}$) of 350 kg m$^{-1}$; (d) modelled water erosion according to reference parameterization ($ktc$: 150; also see Table 1).